# Sequentially learning the topological ordering of directed acyclic graphs with likelihood ratio scores

**Gabriel Ruiz**[*]                                                          *ruizg@ucla.edu*
*Department of Statistics*
*University of California, Los Angeles*

**Oscar Hernan Madrid Padilla**                                    *oscar.madrid@stat.ucla.edu*
*Department of Statistics*
*University of California, Los Angeles*

**Qing Zhou**[†]                                                            *zhou@stat.ucla.edu*
*Department of Statistics*
*University of California, Los Angeles*

**Reviewed on OpenReview:** *https://openreview.net/forum?id=4pCjIGIjrt*

## Abstract

Causal discovery, the learning of causality in a data mining scenario, has been of strong scientific and theoretical interest as a starting point to identify "what causes what?" Contingent on assumptions and a proper learning algorithm, it is sometimes possible to identify and accurately estimate an underlying directed acyclic graph (DAG), as opposed to a Markov equivalence class of graphs that gives ambiguity of causal directions. The focus of this paper is in highlighting the identifiability and estimation of DAGs through a sequential sorting procedure that orders variables one at a time, starting at root nodes, followed by children of the root nodes, and so on until completion. We demonstrate a novel application of this general sequential approach to estimate the topological ordering of the DAG corresponding to a linear structural equation model with a non-Gaussian error distribution family. At each step of the procedure, only simple likelihood ratio scores are calculated on regression residuals to decide the next node to append to the current partial ordering. The computational complexity of our algorithm on a $p$-node problem is $\mathcal{O}(pd)$, where $d$ is the maximum neighborhood size. Under mild assumptions, the population version of our procedure provably identifies a true ordering of the underlying DAG. We provide extensive numerical evidence to demonstrate that this sequential procedure scales to possibly thousands of nodes and works well for high-dimensional data. We accompany these numerical experiments with an application to a single-cell gene expression dataset. Our `R` package with examples and installation instructions can be found at `https://gabriel-ruiz.github.io/scorelingam/`.

## 1 Introduction

With observational data alone, causal inference using an accurate directed acyclic graph (DAG) has been shown to provide results that are up to par with the quintessential randomized controlled experiment (Pearl, 2009; Malinsky et al., 2019). However, it is difficult to imagine that this approach, with its reliance on strong domain knowledge about the system of variables at hand, can be applied to cases with large numbers of variables and little background on how they are all related. For example, specifying a DAG may be difficult in bioinformatics and fields of science where "big data" was previously unavailable and we are now trying to get a grasp of it. On the other hand, causal discovery—learning the DAG structure for Bayesian networks

---

[*]Supported by NSF grant DGE-1650604.
[†]Supported by NSF grant DMS-1952929.

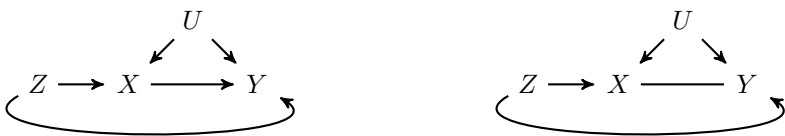

Figure 1: The original DAG (left) and its corresponding CPDAG (right), obtained by keeping the orientation of edges corresponding to the v-structures $Z \to X \leftarrow U$ and $Z \to Y \leftarrow U$, and removing the orientation from all other edges. Note the ambiguity about the causal direction $X \to Y$ vs. $X \leftarrow Y$ in the CPDAG.

from scratch—has its own pitfalls, such as a super-exponentially increasing number of networks in the search space as the number of nodes grows and the fact that it is possible for multiple directed acyclic graphs and the Bayesian networks they encode to map to the same joint distribution–a phenomenon called Markov equivalence (Frydenberg, 1990; Verma & Pearl, 1990; Peters et al., 2017; Verma & Pearl, 2022). Nonetheless, a large effort has been devoted to the structure learning of DAGs as a preliminary data mining step in the scientific pipeline (Stekhoven et al., 2012; Nandy et al., 2017; Malinsky & Spirtes, 2017; Henckel et al., 2022; Lee et al., 2022).

The present paper focuses on a sequential learning algorithm, which we show can be quite scaleable, along with novel identification theory for a specific application. The main task of the approach we advocate for is to sequentially estimate a topological ordering of the DAG, a permutation of node labels such that every parent must precede its children. To help with scalability in practice, we also make use of a priori known neighborhood sets, such as a Markov blanket of a node. In order to demonstrate the theoretical promise of this procedure, we discuss existing identification results that make use of it. We also provide new theory for a linear structural equation model (SEM) first studied in Shimizu et al. (2006). The novelty of our application of the sequential sorting procedure to this SEM compared to the state-of-the-art for it is the scalability of our procedure to a large number of nodes in the underlying graphical model.

Representative methods for causal discovery under the assumption of no unobserved confounding (causal sufficiency) include the Peter-Clark (PC) algorithm (Spirtes & Glymour, 1991) and Greedy Equivalence Search (GES) (Chickering, 2002). The PC algorithm is a constraint-based method due to its use of conditional independence queries, while GES is considered a score-based method for the objective function it seeks to optimize across the space of graphical models. Without additional structural assumptions, the best these methods can generally do in the limit of sample size ($n \to \infty$) is to obtain a Markov equivalence class (MEC) of DAGs, visualized typically by a single Completed Paritially Directed Acyclic Graph (CPDAG) as in Figure 1. Each DAG in the MEC, obtainable by orienting undirected edges in the CPDAG without introducing a cyclic path nor a "v-structure," encodes the same set of *d-separation* relations that imply marginal and conditional independence relations between triplets of variable subsets in their underlying joint distribution (Spirtes & Glymour, 1991).

When additional assumptions are justified, such as strict non-linearity of structural equations, or non-Gaussianity of noise terms in a linear structural equation model, a unique DAG can be identified (Bühlmann et al., 2014; Shimizu et al., 2006). When we are not willing to make the assumption of causal sufficiency, the Fast Causal Inference (FCI) algorithm provides an alternative at the cost of a potentially less precise, though possibly more accurate, graphical model compared to a DAG or CPDAG (Spirtes et al., 2000; Malinsky & Spirtes, 2017). Beyond the case of independent and identically distributed (iid) copies from a distribution that our DAG of interest satisfies the Markov property with respect to, Glymour et al. (2019) and Peters et al. (2017) provide reviews on the trade-offs of different algorithms and what can and cannot be done when there is additional structure, such as the case that the system of variables varies in time. In the context of Earth system sciences, Runge et al. (2019) review causal discovery methods. Structure learning has also been explored for its possibility to explain the black-box nature of state-of-the-art deep learning architectures (Sani et al., 2020). Moreover, Zheng et al. (2018) and its extension to Zheng et al. (2020) provide an approach to optimize a non-convex score function in DAG space by using a smooth characterization of an adjacency matrix's acyclicity constraint.

### 1.1 Review of relevant work

Specific to our task of learning a topological ordering for an underlying acyclic graphical model, we now review some relevant work. Let us first formally define a so-called topological ordering of a DAG, the target parameter we seek to estimate. Let $[m] = \{1, \ldots, m\}$ for integer $m \geq 1$. Moreover, let $PA_k \subseteq [p]$ denote the nodes in the DAG with directed edges going into node $k$.

**Definition 1.1.** A topological ordering for a DAG $\mathcal{G}$ is given by a permutation $\pi : [p] \to [p]$ such that every parent node precedes its child in the ordering:

$$j \in PA_k \implies \pi^{-1}(j) < \pi^{-1}(k).$$

Importantly, we note that the discrete search space across $p!$ permutation functions in search of one that satisfies Definition 1.1 can be quite cumbersome (Raskutti & Uhler, 2018; Solus et al., 2021). Several heuristic score-based methods have been developed to cope with the search space, however, it remains the case that score-based approaches for ordering search are NP hard in general (Chickering, 1996; Ye et al., 2021). Along these lines, recent work by Ye et al. (2021) provides one approach under the case of a linear Bayesian network with Gaussian noise. The similar non-parametric approach of Solus et al. (2021) and Wang et al. (2017) requires a consistent conditional independence testing procedure to decide the presence or absence of an edge in the DAG corresponding to a given $\tilde{\pi}$ in the search space. The empirical results of these approaches are all promising. However, these works do not provide guarantees on whether the search can terminate at a point well before querying all permutations or DAGs in order to achieve optimal (statistically consistent) results.

Complementary to these advances, we here study a simpler approach to estimate a permutation for which the search has a pre-determined number of steps: $\mathcal{O}(pd)$ in the number of least squares residual updates, where $d \leq p$ is the maximum neighborhood size of a node. The objective of our work is to estimate one such permutation $\pi$ from observed data $\mathbf{X} \in \mathbb{R}^{n \times p}$ of sample size $n$. We denote this estimate as $\hat{\pi}$. To do so, we will apply Algorithm 1 for $t = 1, 2, \ldots, p$ until all nodes are sorted. At step $t$, given an input partial ordering $\mathcal{A}_t = (\hat{\pi}(1), \ldots, \hat{\pi}(t-1))$, the algorithm selects a node $\hat{\pi}(t) \notin \mathcal{A}_t$ to append to the estimated ordering by maximizing an appropriate score $\mathcal{S}(k, \mathcal{A}_t; \mathbf{X})$ of interest.

---

**Algorithm 1:** Continue a Topological Ordering

---

**Data:** The partial ordering $\mathcal{A}_t$ and data matrix $\mathbf{X} \in \mathbb{R}^{n \times p}$
**Result:** The continued partial ordering $\mathcal{A}_{t+1}$
$\hat{\pi}(t) \leftarrow \arg\max_{k \notin \mathcal{A}_t} s_k$
$\mathcal{A}_{t+1} \leftarrow \mathcal{A}_t \cup \{\hat{\pi}(t)\}$

---

There exist structure learning methods that use the general approach in Algorithm 1 to sequentially construct a topological ordering. These approaches motivate our present work and include the following. Peters et al. (2014) apply Algorithm 1 under an assumption of strictly nonlinear structural equations with additive noise. Meanwhile, Ghoshal & Honorio (2017), Park (2020), Park & Kim (2020), and Chen et al. (2019) apply this sequential sorting procedure under a bounded conditional variance assumption: $a \leq \mathbb{V}[X_j | X_{PA_j}] \leq b$ for each $j \in [p]$ and some unknown positive constants $a \leq b$ restricted by the signal a parent sends its child node. Notably, Park (2020) contains an extended discussion on the case of a node's possibly non-linear relation with its parents, while Gao et al. (2020) further explore the computational scalability for the sequential application of Algorithm 1 to estimate non-linear structural equation models under this bounded conditional variance assumption. With respect to linear SEMs with non-Gaussian noise, applications of Algorithm 1 include Shimizu et al. (2011), Hyvärinen & Smith (2013), and Wang & Drton (2019), while Zeng et al. (2020) construct the topological ordering in reverse starting with child-less nodes. We believe there exists potential to scale up the estimation of each of these models. We focus on the linear SEM with non-Gaussian noise here.

### 1.2 Paper Contribution and Outline

Our application of Algorithm 1 is to a linear SEM with non-Gaussian noise under causal sufficiency, which is known as the linear non-Gaussian acyclic model (LiNGAM). In terms of theoretical guarantees for the

estimation of LiNGAM, Shimizu et al. (2006) and Shimizu et al. (2011) provide identifiabiliy results for the respective LiNGAM learning procedures–that is, with knowledge of the true distribution defined by the LiNGAM and an oracle for conditional independence queries in the case of latter. Meanwhile Wang & Drton (2019) provide formal statistical consistency results for their LiNGAM-learning procedure.

Although the above methods have very nice theoretical guarantees, their practical application is limited as they do not presently scale well to large graphs, say with thousands of nodes, as confirmed in the numerical results in this paper. Therefore, we develop a fast sequential learning method that can estimate large graphs in practice. At each step of this method, a node is selected to append to a partial ordering, so that after $p$ steps, where $p$ is the number of nodes in the underlying graph, a full ordering of all the nodes will be produced. Compared to the existing works on LiNGAM, the main contributions of our work are:

1. Based on a specified error distribution, we define a novel likelihood ratio score which is used at each step in our sequential algorithm. The evaluation of the likelihood ratio only involves linear regression and residual calculation. There are no tuning parameters.

2. We prove that at the population-level, this sequential algorithm will identify a true ordering of the underlying DAG under proper assumptions on the LiNGAM.

3. Our sequential method is computationally tractable with computational complexity $O(p^2)$ for the number of updates used in the entire algorithm. If prior knowledge on the Markov blankets of the nodes is provided, the computational complexity can be further reduced to $O(pd)$, where $d$ is the maximum size of the Markov blankets. This is in sharp contrast to traditional score-based approaches for ordering search, which are NP hard in general (Chickering, 1996; Ye et al., 2021).

4. The computational complexity of the finite version of our procedure is $\mathcal{O}(pdmn)$ in the number of operations needed. The term $m < p$ is the maximum number of ancestors a node can have (see Algortihm 2. Meanwhile, the procedure in Shimizu et al. (2011) is stated to be $\mathcal{O}(np^3M^2 + p^4M^3)$ in the number of operations, where $M$ is the maximal rank found by the low-rank decomposition used in their kernel-based independence measure of choice. Moreover, Wang & Drton (2019) address computational complexity in passing by stating that "the computational complexity of [their sorting algorithm] is exponential in [the maximum number of parents a node can have], so in practice [the maximum number of parents a node can have] must remain relatively small."

The rest of the paper is organized as follows. In the rest of this section, we formally introduce the linear SEM of interest. Next, in Section 2 we will introduce our approach: §2.2 discusses the conditions for this approach to work; §2.3 provides a formal identifiability result; and §2.4 provides the finite sample version of the algorithm. Section 3 presents simulation results for our procedure for small and large-sized Bayesian networks, along with an application to single-cell gene expression data. Finally, we conclude with a summary of our findings and discussion of future work.

## 1.3   Review of LiNGAM

We follow closely here the definition of a LiNGAM given by Shimizu et al. (2006).

**Definition 1.2.** (Linear Non-Gaussian Acyclic Model)
For $p \geq 2$, let $\mathcal{G}$ be a DAG on $p$ nodes and $\mathbf{B} \in \mathbb{R}^{p \times p}$ be the weighted adjacency matrix of $\mathcal{G}$ such that $\mathbf{B}_{jk} \neq 0$ means $j \in PA_k$, the parent set of node $k$. Let $\epsilon = (\epsilon_1, \ldots, \epsilon_p)$ such that $\epsilon_k \sim g(\cdot; \theta_k)$ independently, with $g(\cdot; \theta)$ a density of a non-Gaussian distribution parameterized by $\theta \in \mathbb{R}^q$, and $\mathbb{E}[\epsilon_j] = 0$ $(j = 1, \ldots, p)$. We say $X \in \mathbb{R}^p$ follows a LiNGAM with DAG $\mathcal{G}$ if

$$X_k = \sum_{j \in PA_k} \mathbf{B}_{jk} X_j + \epsilon_k, \qquad k = 1, \ldots, p. \tag{1}$$

Denote $X$'s probability density function (pdf) as $f(x)$.

The scalar form of the linear SEM in Equation 1 can be rewritten in vector form as $X = \mathbf{B}^T X + \epsilon$. Put $\mathbf{M} = (\mathbb{I}_p - \mathbf{B})^{-T}$, a matrix with ones on its diagonal. Let $AN_k$ denote the ancestor set of node $k$: $a \in AN_k$ means there exists a direct path starting at node $a$ and ending at node $k$, $a \to \cdots \to k$. Then we arrive at $X = \mathbf{M}\epsilon$ in vector form and $X_k = \epsilon_k + \sum_{j \in AN_k} \mathbf{M}_{kj}\epsilon_j$ in scalar form for all $k \in [p]$. Noting that $\mathbf{M}$ serves as a mixing matrix for the independent components in $\epsilon$, we may think of the estimation of this linear SEM as an instance of Independent Component Analysis (ICA) (Hyvärinen & Oja, 2000). Shimizu et al. (2006) discuss the connection between LiNGAM and ICA.

## 2 Methodology and algorithm

In this section, we introduce both the population-level and finite-sample versions of our sorting procedure. We also show that our choice of summary score $\mathcal{S}(k, \mathcal{A}_t; \mathbf{X})$ in Algorithm 1 will lead to the identification of a topological ordering of the true DAG $\mathcal{G}$ used to define the linear SEM of Definition 1.2. We first go over some useful notation.

### 2.1 Summary of Notation

For a positive integer $m$, we write $[m] = \{1, 2, \ldots, m\}$. For any set $S$, $|S|$ will denote its cardinality: the number of unique elements it contains. $|S| = 0$ means $S = \emptyset$, the set with no elements. For sets $T \subseteq [m], S \subseteq [r]$, and matrix $A \in \mathbb{R}^{m \times r}$, $A_{\cdot S} \in \mathbb{R}^{m \times |S|}$ is the sub-matrix defined by indexing columns $S$ in $A$, while $A_{T \cdot} \in \mathbb{R}^{|T| \times r}$ is the sub-matrix given by indexing rows $T$ in $A$. Similarly, $A_{TS} \in \mathbb{R}^{|T| \times |S|}$ is the sub-matrix indexing rows $T$ and columns $S$ of $A$. Similarly, for a vector $v \in \mathbb{R}^m$, we will write $v_T \in \mathbb{R}^{|T|}$ to denote the subset of entries indexed by $T$. We will also sometimes write $(v_j; j \in T)$ to denote $v_T$. When the indexing sets are singletons, $S = \{k\}$ or $T = \{j\}$, we will simply write $A_{\cdot k}, A_{j\cdot}, A_{jk}$, and $v_k$, respectively. For two sets $S$ and $T$, we will make use of set operations such as their intersection: $S \cap T = \{a : a \in S \text{ and } a \in T\}$; their union: $S \cup T = \{a : a \in S \text{ or } a \in T\}$; and their difference: $S \backslash T = \{a : a \in S \text{ and } a \notin T\}$. For sets $S_1, \ldots, S_K$, we denote their intersection as $\bigcap_{j=1}^K S_j$ and their union as $\bigcup_{j=1}^K S_j$.

We will refer to a directed acyclic graph by $\mathcal{G} = (\mathcal{V}, \mathcal{E})$. $\mathcal{G}$ is a collection of singular vertices, $\mathcal{V} = [p]$, and vertex pairs, $\mathcal{E} \subseteq V \times V$. Specifically, $(j, k) \in \mathcal{E}$ means that $j$ has a directed edge into $k$ in $\mathcal{G}$. $PA_k$ denotes the parent set of node $k \in \mathcal{V}$: $j \in PA_k$ only if $(j, k) \in \mathcal{E}$. The set of all of node $j$'s children, $CH_j$, contains all nodes $k$ such that $(j, k) \in \mathcal{E}$. Let $\widehat{N}_k \subseteq \mathcal{V}$ denote the set of nodes that are thought to be most predictive of node $k$, which we will refer to as node $k$'s neighborhood. Let $\widehat{N}_{kt} := \widehat{N}_k \cap \mathcal{A}_t$, which is the subset of the neighborhood set that has been ordered at step $t$ of our procedure (Algorithm 1). For the cases where $|\widehat{N}_{kt}| \geq 1$, we will make use of least squares residuals for calculating the score $\mathcal{S}(k, \mathcal{A}_t)$. At the population-level, the residual is

$$R_{kt} := \begin{cases} X_k & \text{if } |\widehat{N}_{kt}| = 0 \\ X_k - \beta_{kt}^T X_{\widehat{N}_{kt}} & \text{otherwise} \end{cases}, \tag{2}$$

where $\beta_{kt}$ is the least-squares regression coefficient vector,

$$\beta_{kt} = \left( \mathbb{E}\left[ X_{\widehat{N}_{kt}} X_{\widehat{N}_{kt}}^T \right] \right)^{-1} \mathbb{E}\left[ X_{\widehat{N}_{kt}} X_k \right].$$

Also,

$$\eta_{kt} := \arg\max_{\eta} \mathbb{E}_{X \sim f(x)} \left[ \log g(R_{kt}; \eta) \right],$$

while $\phi(r_{kt}; \sigma_{kt})$ is the density for $\mathcal{N}\left(0, \sigma_{kt}^2 = \mathbb{V}[R_{kt}]\right)$, i.e. the normal density that matches the mean and variance of $R_{kt}$. The corresponding sample analogues of $R_{kt}, \beta_{kt}, \eta_{kt}$ and $\sigma_{kt}$, are discussed in §2.4.

### 2.2 Assumptions

Our main assumptions are on the distributions of the independent entries of the error vector $\epsilon$. We consider restricting our class of densities $\{g(\cdot; \theta_k)\}_{1 \leq k \leq p}$ for the noise terms in Definition 1.2 to a scale-location family

in which the $\theta_k > 0$ are the scale parameters, such as the Laplace family of distributions, the Logistic family of distributions, or a Scaled-t distribution family (same degrees of freedom). This is summarized in Assumption 2.1.

**Assumption 2.1.**
Let $U \sim g(\cdot; \theta_0)$ with $\theta_0 > 0$ and $\mathbb{E}[U] = 0$. For each $k = 1, 2, \ldots, p$, the density of the error $\epsilon_k$ satisfies

$$g(e; \theta_k) = \frac{\theta_0}{\theta_k} g(\theta_0 e / \theta_k; \theta_0).$$

That is, $\epsilon_k \overset{d}{=} (\theta_k / \theta_0) U$, an equality in distribution.

Our next assumption for the linear SEM of interest is on linear combinations of the noise terms. This condition is related to Lemma 2.6 in § 2.3.1, a key result about how to characterize the regression residuals of Equation 2 as linear combinations of "independent components."

**Assumption 2.2.** For any $j = 1, 2, \ldots, p$ and any $a \in \mathbb{R}^p$ with at least two non-zero entries, the linear combination $a^T \epsilon$ does not follow the same distribution as $\epsilon_j$.

Notable disagreements with Assumption 2.2 are when the $\epsilon_j$ are all Gaussian distributed (not the case for LiNGAM), or when the $\epsilon_j$ are all Poisson-distributed. Notable agreements with Assumption 2.2 (and Assumption 2.1) are the cases where the $\epsilon_j$ are all Laplace-distributed, all Logistic-distributed, or all Scaled-t distributed (same degrees of freedom). More generally, we may start with any reference distribution, then specify the distribution for error terms in Definition 1.2 as re-scalings of the original reference distribution. So long as the characteristic function (Forbes et al., 2010) for the sum of any two independent random variables following a distribution in this scale-location family does not have the same form as the characteristic function of the original distribution of a summand, then Assumption 2.1 and Assumption 2.2 are satisfied.

To allow for a quicker sorting procedure in practice, we may make use of an *a priori* known support set for the neighborhood of each node in the DAG. We consider these neighborhood sets to arise based on domain knowledge, previous studies, or a pre-processing step such as with neighborhood lasso regression (Meinshausen & Bühlmann, 2006). We highlight this usage in Assumption 2.3.

**Assumption 2.3.** For node $k$, denote its neighborhood estimate as $\widehat{N}_k$. Assume for each $k = 1, 2, \ldots, p$ that:

$$\widehat{N}_k \supseteq MB_k := PA_k \cup CH_k \cup \bigcup_{j \in CH_k} PA_j \backslash \{k\},$$

where $MB_k$ is known as the Markov Blanket of node $k$: the set of its parents $PA_k$, its children $CH_k$, and its co-parents $\bigcup_{j \in CH_k} PA_j \backslash \{k\}$.

*Remark* 2.4. When we consider the population-level version of our algorithm in this section (i.e. we have infinite $n$), we can take $\widehat{N}_k = [p] \backslash \{k\}$ for each $k$ so that Assumption 2.3 holds trivially. For the finite sample version of our procedure discussed in Section 2.4, we will make use of Ordinary Least Squares (OLS) linear regressions which require the design matrix to be of full column rank. So if $p \ll n$, we may also take $\widehat{N}_k = [p] \backslash \{k\}$ for each $k$. In the case that $p \gg n$ or $p \approx n$, sparse neighborhood sets, $|\widehat{N}_k| \ll n$, will reduce the number of covariates in the required OLS regressions.

## 2.3 Our Choice of a Likelihood Ratio Score

In Algorithm 1, we will select the next node to continue our constructed topological ordering as:

$$\hat{\pi}(t) = \arg\max_{k \notin \mathcal{A}_t} \mathbb{E}_{X \sim f(x)} \left[ \log \frac{g(R_{kt}; \eta_{kt})}{\phi(R_{kt}; \sigma_{kt})} \right]. \tag{3}$$

Here, $\mathbb{E}_{X \sim f(x)}[\cdot]$ denotes expectation with respect to $X$'s true density, $f(x)$ in Definition 1.2.

Take a "valid" node to continue the ordering $\mathcal{A}_t$ as a node whose parent nodes are all contained in the set $\mathcal{A}_t$. And take an "invalid" node to continue the ordering $\mathcal{A}_t$ as a node whose parent nodes are not all in the set

$\mathcal{A}_t$. The log-likelihood ratio in Equation 3 can be thought of as a score that allows us to distinguish between valid and invalid nodes by telling us "how non-Gaussian" the residual $R_{kt}$ is. If the residual is explained by a Gaussian distribution well relative to the non-Gaussian distribution in the assumed family, then we expect the log-likelihood ratio to be smaller. Otherwise, if the Gaussian density is not a good fit relative to $g(r_{kt}; \eta_{kt})$, then we have stronger evidence to believe that node $k$ is a valid node to continue the ordering.

In Theorem 2.5, we claim that using Equation 3 leads to the identification of a valid topological ordering–our main result. The proof of Theorem 2.5 is in Appendix A.1. Key to the proof, we note that Equation 3 can also be written equivalently using the difference of two KL-divergence (Cover & Thomas, 2005) terms:

$$\hat{\pi}(t) = \arg\max_{k \notin \mathcal{A}_t}\{D_{KL}\left(f_{kt}(r_{kt})||\phi(r_{kt}; \sigma_{kt})\right) - D_{KL}\left(f_{kt}(r_{kt})||g_k(r_{kt}; \eta_{kt})\right)\}. \tag{4}$$

Here, $f_{kt}(r_{kt})$ is the probability density function of the residual $R_{kt}$, which in general can be different from the specified density $g(r_{kt}; \eta_{kt})$. The first term in Equation 4 deals with the divergence of the $R_{kt}$'s true distribution from a Gaussian distribution. This term will be largest when $k$ is a valid node to append to the current ordering. The second term in Equation 4 concerns the deviation between $R_{kt}$'s true distribution and the specified distribution given by the probability density function $g(r_{kt}; \eta_{kt})$. It will be non-zero when node $k$ is an invalid node to append to the current ordering. Section 2.3.1 elaborates on this explanation by introducing the key lemmas for the proof of Theorem 2.5.

**Theorem 2.5.** *Let $X \in \mathbb{R}^p$ follow a LiNGAM with DAG $\mathcal{G}$. If Assumptions 2.1, 2.2 and 2.3 hold, then applying Algorithm 1 at all steps $t = 1, 2, \ldots, p$ with the score*

$$\mathcal{S}(k, \mathcal{A}_t) = \mathbb{E}_{X \sim f(x)}\left[\log \frac{g(R_{kt}; \eta_{kt})}{\phi(r_{kt}; \sigma_{kt})}\right]$$

*will identify a permutation $\hat{\pi} = (\hat{\pi}(1), \ldots, \hat{\pi}(p))$ that is a topological ordering of $\mathcal{G}$.*

Theorem 2.5 suggests that the maximization at each iteration in which we apply Algorithm 1 can be done easily. This differs from maximizing a score over a whole ordering which may also lead to identification of the true MEC, but is in general NP hard (not tractable). Relatedly, Section 2.3.2 gives additional motivation for the choice of $\mathcal{S}(k, \mathcal{A}_t)$ in Theorem 2.5 as one that allows us to greedily optimize the mean log-likelihood when the full ordering is only partially discovered.

### 2.3.1 Key Lemmas for the proof of Theorem 2.5

The formal proof of Theorem 2.5 in § A.1 below is an inductive application of the following reasoning when applying Algorithm 1 at a given step $t$. Lemma 2.6 suggests that invalid nodes' residuals, $R_{kt}$, are a linear combination of two or more entries in the vector $\epsilon$, while for valid nodes $\ell$ we have $R_{\ell t} = \epsilon_\ell$. Under Assumption 2.2, this means that the term $D_{KL}\left(f_{kt}(r_{kt})||g_k(r_{kt}; \eta_{kt})\right)$ in Equation 4 will be zero only if node $k$ is valid to continue the ordering at step $t$.

**Lemma 2.6** (Characterizing nodes' residuals as linear combinations of independent components)**.**
*Assume that $\mathcal{A}_t$ is correct so far in the sense that for each $a \in \mathcal{A}_t$, we have $PA_a \subseteq \mathcal{A}_t$. Also assume Assumption 2.3 holds. We have that:*

- *If $k \in S_t$ is a valid node to continue the ordering, i.e. $PA_k \subseteq \mathcal{A}_t$, then:*

$$R_{kt} = X_k - \beta_{kt}^T X_{\widehat{N}_{kt}} = \epsilon_k.$$

- *Otherwise, if $k$ is not a valid node, then $R_{kt}$ is a linear combination of more than one independent component in $\epsilon$.*

The natural follow up question is what the behavior is for the term $D_{KL}\left(f_{kt}(r_{kt})||\phi(r_{kt}; \sigma_{kt})\right)$ in Equation 4 when $k$ is valid vs. invalid to continue the ordering. Lemma 2.7 provides this insight: for valid nodes to continue an ordering, this term's value is no less than the same term's value for invalid nodes. Given this and that the term $D_{KL}\left(f_{kt}(r_{kt})||g_k(r_{kt}; \eta_{kt})\right)$ in Equation 4 will be zero only if node $k$ is valid to continue the ordering at step $t$, Equation 4 (equivalently Equation 3) helps us select a valid node to continue the topological ordering at step $t$.

**Lemma 2.7** (KL Divergence from Gaussianity for valid and invalid nodes' residuals).
*Let $X \in \mathbb{R}^p$ be a LiNGAM from Definition 1.2 that satisfies Assumptions 2.1 and 2.3. Assume that $\mathcal{A}_t$ is correct in the sense that $PA_a \subseteq \mathcal{A}_t$ for all $a \in \mathcal{A}_t$. Let $k \in [p]\backslash\mathcal{A}_t$ be an invalid node to continue the ordering in the sense that there exists $j \in PA_k$ such that $j \in [p]\backslash\mathcal{A}_t$. And let $\ell \in [p]\backslash\mathcal{A}_t$ be a valid node to continue the ordering in the sense that $PA(\ell) \subseteq \mathcal{A}_t$. Then the least squares residual $R_{\ell t} \sim f_{\ell t}(r_{\ell t})$ is no closer to Gaussian than $R_{kt} \sim f_{kt}(r_{kt})$ in the sense that:*

$$D_{KL}\left(f_{kt}(r_{kt})||\phi(r_{kt}; \sigma_{kt})\right) \leq D_{KL}\left(f_{\ell t}(r_{\ell t})||\phi_{\ell t}(r_{\ell t})\right), \tag{5}$$

*where $\phi_{kt}$ and $\phi_{\ell t}$ are the respective densities for*

$$\tilde{R}_{kt} \sim \mathcal{N}(\mathbb{E}[R_{kt}], \mathbb{V}[R_{kt}]) \text{ and } \tilde{R}_{\ell t} \sim \mathcal{N}(\mathbb{E}[R_{\ell t}], \mathbb{V}[R_{\ell t}]).$$

In light of Lemma 2.6, Lemma 2.7 makes sense under a Central Limit Theorem-like argument: a sum of two or more random variables is closer to Gaussian than each summand alone. Of particular note, a key result that helps show why Lemma 2.7 holds is *Theorem 17.8.1* of Cover & Thomas (2005), a restatement of the entropy-power inequality. This restatement says that the differential entropy for a sum of any two independent random variables, $U$ and $V$, is no less than the differential entropy for the sum of two strategically defined Gaussian random variables, each having the same differential entropy as $U$ and $V$ (rather than the same first two moments), respectively.

### 2.3.2 Greedy Choice of a Factor to Optimize the Joint Likelihood Function

We now give further motivation to the choice of the mean log-likelihood ratio score in Equation 3. Let vector $X \sim f(x)$, where $f(x)$ is the density induced by a LiNGAM as in Definition 1.2. Consider $X$'s expected log-likelihood as a function of the permutation $\pi$:

$$\mathcal{L}(\pi) = \sum_{j=1}^{p} \mathbb{E}_{X\sim f(x)}\left[\log g\left(X_j - [\mathbf{B}_{\cdot j}^{\pi}]^T X; \theta_j^{\pi}\right)\right], \tag{6}$$

where $\theta_j^{\pi}$ is the corresponding scale parameter according to Assumption 2.1. Here, $\mathbf{B}^{\pi}$ is the acyclic weighted adjacency matrix that arises from a population-level least squares objective such that the $\pi(j)$-th column is given by:

$$\mathbf{B}_{\cdot\pi(j)}^{\pi} = \arg\min_{\substack{\gamma\in\mathbb{R}^{p\times 1}: \ \gamma_k=0 \ \forall k \\ \text{s.t } \pi^{-1}(k) \ \geq \ j}} \mathbb{E}_{X\sim f(x)}\left[(X_{\pi(j)} - \gamma^T X)^2\right].$$

That is, the column $\mathbf{B}_{\cdot\pi(j)}^{\pi}$ is comprised of the least squares coefficients when linearly regressing $X_{\pi(j)}$ onto its predecessors, if any, in the ordering given by $\pi$. Now let $\phi_j^{\pi}$ be the density for the Gaussian distribution having the same first two moments as:

$$R_j^{\pi} := X_j - [\mathbf{B}_{\cdot j}^{\pi}]^T X.$$

Define

$$\tilde{\mathcal{L}}(\pi) := \sum_{j=1}^{p} \mathbb{E}_{X\sim f(x)}\left[\log \phi_j^{\pi}\left(R_j^{\pi}\right)\right] \text{ and } \kappa := \mathbb{E}_{X\sim f(x)}\left[\log \mathcal{N}(X; 0, \mathbb{V}[X])\right].$$

Here, $\mathcal{N}(x; \mathbb{E}[X], \mathbb{V}[X])$ denotes the density for a $p$-variate Gaussian distribution with the same first and second order moments as $X$. Due to the relation between $\mathbf{B}^{\pi}$ and the generalized Cholesky factorization of $\mathbb{V}[X]$, Ye et al. (2021) shows that we actually have the equality:

$$\tilde{\mathcal{L}}(\pi) = \kappa, \quad \text{for all } \pi. \tag{7}$$

The term $\kappa$ is notably a constant that does not depend on $\pi$. Thus, maximizing Equation 6 with respect to $\pi$ is the same as maximizing the expected log-likelihood ratio given by:

$$(\mathcal{L} - \tilde{\mathcal{L}})(\pi) = \sum_{j=1}^{p} \mathbb{E}_{X\sim f(x)}\left[\log \frac{g\left(R_j^{\pi}; \theta_j^{\pi}\right)}{\phi_j^{\pi}\left(R_j^{\pi}\right)}\right] = \mathcal{L}(\pi) - \kappa. \tag{8}$$

With all this in mind, we can think of our choice of a node to append to the ordering $\mathcal{A}_t$ at step $t$ as greedily choosing the largest summand,

$$\mathbb{E}_{X \sim f(x)} \left[ \log \frac{g\left(R_{\hat{\pi}(t)}^{\hat{\pi}}; \theta_{\hat{\pi}(t)}^{\hat{\pi}}\right)}{\phi_{\hat{\pi}(t)}^{\hat{\pi}}\left(R_{\hat{\pi}(t)}^{\hat{\pi}}\right)} \right],$$

to add to the known log-likelihood ratio at step $t$:

$$(\mathcal{L} - \tilde{\mathcal{L}})_t(\hat{\pi}) := \begin{cases} 0 & t = 1 \\ \sum_{j=1}^{t-1} \mathbb{E}_{X \sim f(x)} \left[ \log \frac{g\left(R_{\hat{\pi}(j)}^{\hat{\pi}}; \theta_{\hat{\pi}(j)}^{\hat{\pi}}\right)}{\phi_{\hat{\pi}(j)}^{\hat{\pi}}\left(R_{\hat{\pi}(j)}^{\hat{\pi}}\right)} \right] & 2 \le t \le p+1 \end{cases}.$$

That is, our sequential application of Algorithm 1 is attempting to greedily maximize Equation 8 one summand at a time.

### 2.4 Finite Sample Sorting Procedure

Assume that we have a data matrix $\mathbf{X} \in \mathbb{R}^{n \times p}$ such that $\mathbf{X}_{i.}$, the $i$-th row, is iid across $i = 1, 2, \ldots, n$ from a distribution defined by a LiNGAM satisfying Assumptions 2.1 and 2.2. Also let Assumption 2.3 hold.

Analogous to the population version of our sorting procedure in Section 2.3, consider:

$$\hat{\beta}_{kt} = \left( \mathbf{X}_{\cdot \widehat{N}_{kt}}^T \mathbf{X}_{\cdot \widehat{N}_{kt}} \right)^{-1} \mathbf{X}_{\cdot \widehat{N}_{kt}}^T \mathbf{X}_{\cdot k} \in \mathbb{R}^{|\widehat{N}_{kt}| \times 1},$$

which exists so long as $1 \le |\widehat{N}_{kt}| \le n$ and $\mathbf{X}_{\cdot \widehat{N}_{kt}}$ is of full column rank almost surely. Further, we define $\hat{R}_{kt} \in \mathbb{R}^{n \times 1}$ as

$$\hat{R}_{kt} = \begin{cases} \mathbf{X}_{\cdot k} & \text{if } |\widehat{N}_{kt}| = 0 \\ \mathbf{X}_{\cdot k} - \mathbf{X}_{\cdot \widehat{N}_{kt}} \hat{\beta}_{kt} & \text{if } |\widehat{N}_{kt}| \ge 1 \end{cases},$$

the vector of residuals which we will use to estimate the pertinent scale parameter of Equation 3, denoted as $\hat{\eta}_{kt}$ and $\hat{\sigma}_{kt}$, respectively. Explicitly, we select the next node to continue an ordering using the empirical analogue of the mean log-likelihood ratio in Equation 3:

$$\hat{\pi}(t) = \arg \max_{k \notin \mathcal{A}_t} \frac{1}{n} \sum_{i=1}^{n} \log \frac{g(\hat{R}_{i,kt}; \hat{\eta}_{kt})}{\phi(\hat{R}_{i,kt}; \hat{\sigma}_{kt})}, \tag{9}$$

where $\hat{R}_{i,kt}$ is the $i$-th entry of the vector $\hat{R}_{kt}$, while $\hat{\sigma}_{kt}^2 := \frac{1}{n} \|\hat{R}_{kt}\|_2^2$ and $\hat{\eta}_{kt} := \arg \max_\eta \sum_{i=1}^{n} \log g(\hat{R}_{i,kt}; \eta)$. For example, if $\eta_{kt}$ is the scale parameter for a Laplace distribution, it can be seen that $\hat{\eta}_{kt} = \frac{1}{n} \|\hat{R}_{kt}\|_1$. In this case, Equation 9 is equivalent to

$$\hat{\pi}(t) = \arg \max_{k \notin \mathcal{A}_t} \log \frac{\hat{\sigma}_{kt}}{\hat{\eta}_{kt}} = \arg \max_{k \notin \mathcal{A}_t} \frac{\|\hat{R}_{kt}\|_2}{\|\hat{R}_{kt}\|_1}. \tag{10}$$

*Remark* 2.8. The Laplace update of Equation 10 exemplifies how simple the maximization of our likelihood ratio score is. After the regression of each unsorted node $X_k$, $k \notin \mathcal{A}_t$, onto $\widehat{N}_{kt}$, we only need to compare the ratio between the two norms of the residual vector $\hat{R}_{kt}$ across unsorted nodes to find $\hat{\pi}(t)$.

Algorithm 2 shows the pseudo-code for the sorting procedure we use in practice, with a strategic update of regression residuals using partial regression that greatly reduces the computation cost. The computational complexity for Algorithm 2 is $\mathcal{O}(pdmn)$ in the number of scalar to scalar multiplication, division, addition, or difference operations. Here, $m < p$ is the maximum number of ancestors a node can have, which is a bound on the iterations for the inner-most for loop in Algorithm 2.

We have also provided the details on the estimation of the scale parameters for Logistic and Scaled-t distributions in Appendix C.

---

**Algorithm 2:** The sorting procedure in practice

---

**Data:** $\mathbf{X} \in \mathbb{R}^{n \times p}$ (standardized), $\{\widehat{N}_k\}_{k=1}^p$
**Result:** $\hat{\pi}(1), \hat{\pi}(2), \ldots, \hat{\pi}(p)$
\# initialize mixing matrix
$\mathbf{M} \leftarrow \mathbb{I}_{p \times p}$
\# initialize residual matrix
$\mathbf{R} \leftarrow \mathbf{X}$
\# initialize scores
$s_k \leftarrow \mathcal{S}(k; \mathbf{R}), k = 1, 2, \ldots, p.$
\# sort the nodes
**for** $t = 1, 2, \ldots, p + 1$ **do**
    $\hat{\pi}(t) \leftarrow \arg\max_{k \notin \mathcal{A}_t} s_k$
    \# update residuals for neighbors of selected node.
    **for** $k \in \widehat{N}_{\hat{\pi}(t)} \backslash \mathcal{A}_t$ **do**
        \# update residuals with partial regression.
        **for** $a \in \{j : \mathbf{M}_{\hat{\pi}(t)j} \neq 0, \mathbf{M}_{kj} = 0\}$ **do**
            $\mathbf{M}_{ka} \leftarrow (\mathbf{R}_{\cdot a}^T \mathbf{R}_{\cdot a})^{-1} \mathbf{R}_{\cdot a}^T \mathbf{R}_{\cdot k}$
            $\mathbf{R}_{\cdot k} \leftarrow \mathbf{R}_{\cdot k} - \mathbf{M}_{ka} \mathbf{R}_{\cdot a}$
        **end**
        \# update score
        $s_k \leftarrow \mathcal{S}(k, \mathcal{A}_t; \mathbf{R})$
    **end**
**end**

---

## 3 Empirical Results

We now present empirical results for synthetic and real-data scenarios. For the boxplots of empirically observed distributions in Figures 2, 3, and 5-7, the three horizontal lines on each box indicate the first, second, and third quartile, respectively, while the vertical lines indicate the range of the empirical distribution.

### 3.1 Simulations on Small Networks

We now present simulation results for networks that are on the smaller end with $35 \leq p \leq 223$. These architectures are downloaded from the `bnlearn.com` Bayesian network repository. We compared our sorting procedure to other LiNGAM learning procedures. Due to their readily available code, the algorithms of interest are "DirectLiNGAM" (Shimizu et al., 2011), "HighDimLingam" (Wang & Drton, 2019), and "ScoreLiNGAM" (our procedure). For each simulation setting, we conduct 30 replicates.

For each choice of $\mathcal{G}$ underlying a LiNGAM, our synthetic data generation schema was as follows. We generated $\mathbf{B}_{jk} \overset{\text{i.i.d.}}{\sim} \text{Uniform}[-0.9, -0.4] \cup [0.4, 0.9]$ for each $(j, k)$ such that $j \in PA_k$, and otherwise set $\mathbf{B}_{jk} = 0$. We generated $\theta_k \overset{\text{i.i.d.}}{\sim} \text{Uniform}[0.4, 0.7]$ across $1 \leq k \leq p$, where $\theta_k$ is the scale parameter for the error distributions as in Assumption 2.1. This specific range of simulation parameters for the weighted adjacency matrix entries and scale parameters is chosen so that the marginal variance of $X_j$ does not explode when $j$ is a node with many ancestors. Finally, we varied sample size as $n = 0.5p, p, 2p, 10p, 50p$. Note that $n = 0.5p$ and $n = p$ represent the high-dimensional setting ($p \geq n$).

Next, a dataset $\mathbf{X} \in \mathbb{R}^{n \times p}$ of iid samples is drawn from the distribution given by the LiNGAM parameterized by $(\mathbf{B}, \theta_1, \ldots, \theta_p)$ and having errors $\epsilon_k \sim g(\cdot; \theta_k)$ across $k \in [p]$. Moreover, we varied the family of the densities $g$ in Assumption 2.1 to be the Laplace, the Logistic, or the Scaled-t distribution (10 degrees of freedom) scale-location families. Finally, ScoreLiNGAM and HighDimLiNGAM were run with knowledge of the true Markov blanket for each node, while DirectLiNGAM was not as it does not have this option. Afterward, the data matrix $\mathbf{X}$ was standardized so that each column has sample standard deviation equal to 1 and sample mean equal to 0.

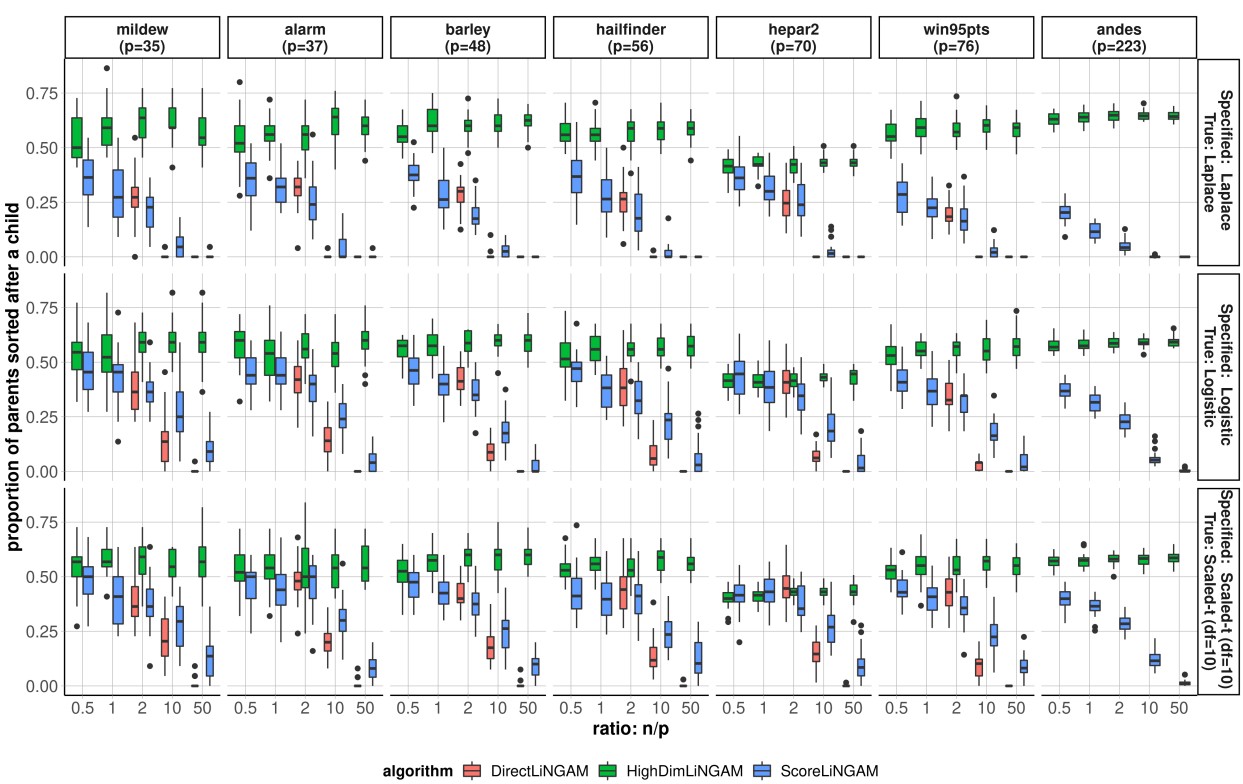

Figure 2: The simulation results comparing LiNGAM estimation procedures. Color indicates sorting procedure. The y axes in each plot indicate sorting estimation error according to Equation 11. The columns of the grid of plots corresponds to the `bnlearn.com` network architecture used. The rows indicate the true and specified scale-location family.

Figure 2 reports the results in terms of order estimation error (lower is better), which we define as:

$$\text{Err}(\hat{\pi}) := \frac{1}{|\mathcal{E}|} \sum_{(j,k) \in \mathcal{E}} \mathbf{1}\{\hat{\pi}^{-1}(j) > \hat{\pi}^{-1}(k)\}. \tag{11}$$

Recall from § 2.1 that $\mathcal{E}$ is the edge set of the true DAG $\mathcal{G}$. The indicator $\mathbf{1}\{\hat{\pi}^{-1}(j) > \hat{\pi}^{-1}(k)\}$ in Equation 11 states whether the ordering with $\hat{\pi}$ erroneously placed parent node $j$ after child node $k$. The normalizing constant

$$|\mathcal{E}| = \sum_{j=1}^{p} \sum_{k=1}^{p} \mathbf{1}\{\mathbf{B}_{jk} \neq 0\},$$

the total number of directed edges in $\mathcal{G}$, gives the order estimation error $\text{Err}(\hat{\pi})$ a value between 0 and 1. Given that the target parameter of our work is a topological ordering as in Definition 1.1, the error metric in Equation 11 concisely captures the extent of a violation to this definition. Each edge $j \rightarrow k$ in the true DAG imposes an independent constraint on a topological ordering. In this regard, $\text{Err}(\hat{\pi})$ is the fraction of these constraints not satisfied by the estimated ordering $\hat{\pi}$. If a node with many parents is placed at the start of an estimated ordering before many of those parents, then this error metric will penalize proportionally. Likewise, this penalty will occur proportionally if a node with many children nodes is sorted after many of those children.

Our ScoreLiNGAM achieved the highest accuracy for all high-dimensional settings ($n \leq p$). DirectLiNGAM became quite comparable until the sample size increased to $n = 2p$ and did a bit better than ScoreLiNGAM

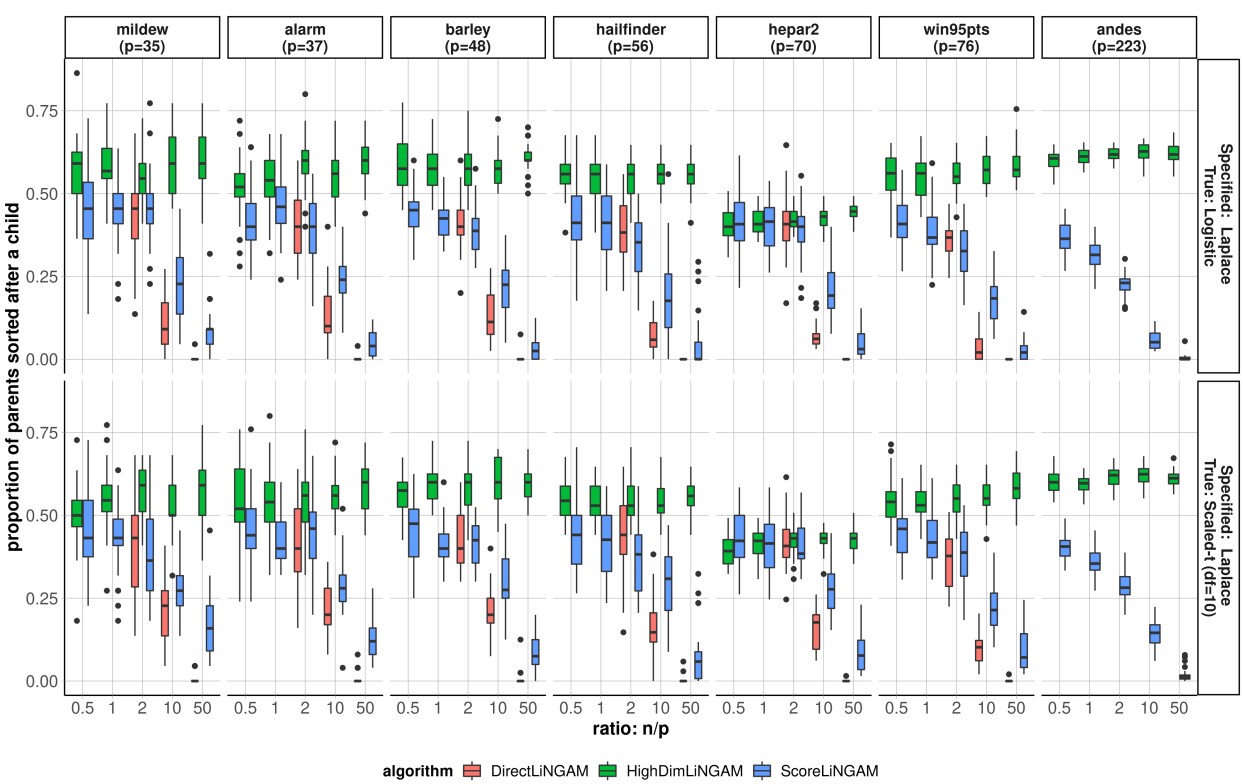

Figure 3: An extension of the simulation results in Figure 2 comparing LiNGAM estimation procedures. Here, we mis-specify the noise distribution for ScoreLiNGAM.

when $n \geq 10p$ (large sample size cases). Note that results are not presented for DirectLiNGAM when $n = 0.5p$ nor $n = p$, because it is not applicable for $n \leq p$. For the Andes network, results for DirectLiNGAM are also not presented as this procedure takes about 118 minutes for a single replicate, which adds up across 90 total replicates. On the other hand, HighDimLiNGAM is generally the least accurate algorithm across all networks and sample sizes. Recall that the data matrix $\mathbf{X}$ is re-scaled. The lack of accuracy change for HighDimLiNGAM is likely owed to the fact that this procedure is not invariant to a re-scaling of the data, as ScoreLiNGAM and DirectLiNGAM are. We also note an apparent good property of DirectLiGAM: it does not require the MB as input, yet it performs well in terms of accuracy at large values of $n$.

We also compared the three methods when the error distributions were mis-specified for ScoreLiNGAM in Figure 3. The true error distributions were Logistic or Scaled-t, but we still used the Laplace update of Equation 10 in ScoreLiNGAM. It is seen that its accuracy was comparable to the result when we correctly specified the error distributions (the other three columns), suggesting some robustness of our method to mis-specification in the error distribution family.

In terms of speed, Figure 4 summarizes this for the win95pts network. The advantage of our method is speed, with our method being no less than 100 times faster the next fastest method. Note: HighDimLiNGAM's procedure is parallelized across 7 threads. For the setting with $n/p = 0.5$ on the win95pts network, ScoreLiNGAM is more than 1000 times faster than HighDimLiNGAM. Appendix B contains details about the implementation of each procedure, along with the machine used to run these experiments. Moreover, Figure 8 in Appendix B contains sorting times for all the settings we considered; there is a similar pattern of ScoreLiNGAM's capability in terms of speed compared to the other methods.

In Appendix B.3, we present results on the barley, alarm, and mildew networks for the case that the data matrix is not re-scaled as the results of this subsection. These results support our claim that HighDimLiNGAM

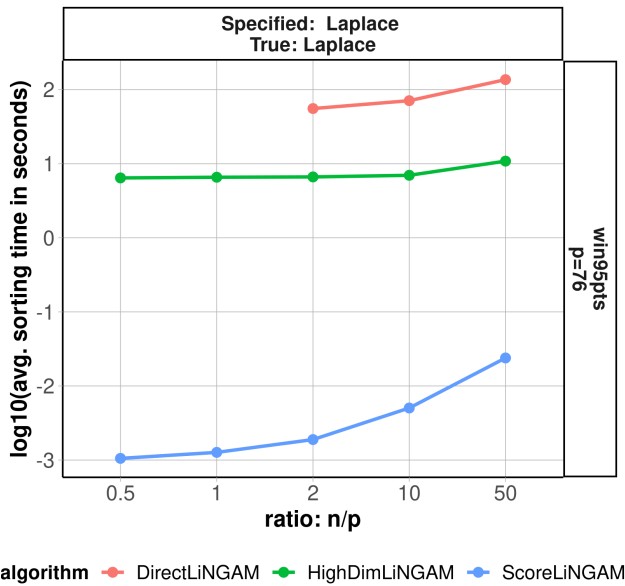

Figure 4: The $\log_{10}$(avg. sorting time in seconds) scale for the various methods applied to the win95pts network.

is sensitive to re-scaling, while DirectLiNGAM and ScoreLiNGAM are not. Next, Appendix B.4 obtains results on these same three networks for the case that neighborhood sets must be estimated. These results help us understand the sensitivity of our method to proper neighborhood specification.

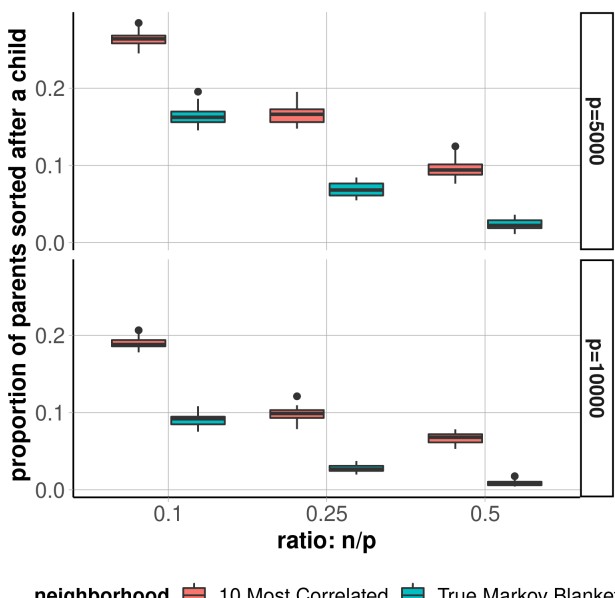

Figure 5: Sorting errors for ScoreLiNGAM under $p = 5000, 10000$ and $n = 0.1p, 0.25p, 0.5p$. Color indicates how the neighborhood sets are constructed.

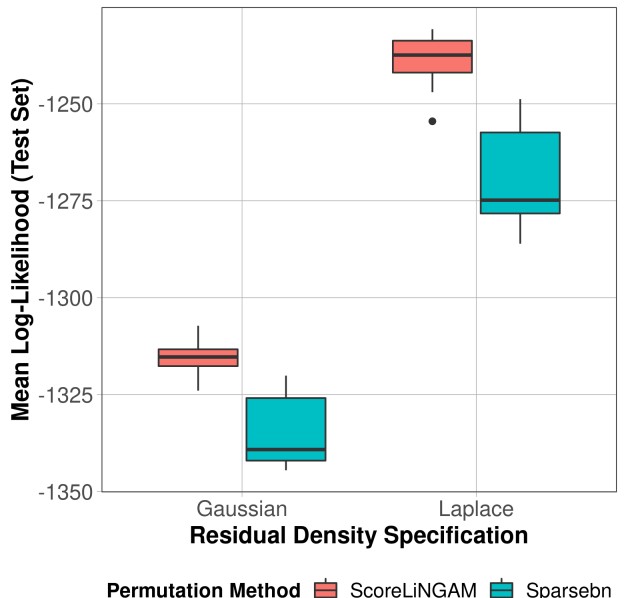

Figure 6: The mean log-likelihood on 1,000 genes for a subset of cells in the data of (Yao et al., 2021), across 50 repetitions.

## 3.2 Larger Network Results

We also simulated large networks with $p = 5000, 10000$ and $n = 0.1p, 0.25p, 0.5p$ to further demonstrate the scalability of ScoreLiGAM. We do not include results in these settings for DirectLiNGAM nor High-DimLiNGAM as they would take too long to run. The network generation is such that 5% of nodes are root nodes (no parents), and all other nodes have between 1 and 2 parents (with equi-probability) which are selected at random from the set of predecessors in a randomly generated permutation. Moreover, $\mathbf{B}_{jk} \overset{\text{i.i.d.}}{\sim} \text{Uniform}[-0.9, -0.4] \cup [0.4, 0.9]$ across $(j, k)$ such that $j \in PA_k$, while $\theta_k \overset{\text{i.i.d.}}{\sim} \text{Uniform}[0.25, 0.9]$ across $1 \le k \le p$ is the scale parameter for the Laplace noise in the synthetic LiNGAM. A new LiNGAM is generated according to this schema for each data replicate.

Figure 5 presents simulation results for ScoreLiNGAM with two different a priori known neighborhood sets. "True Markov Blanket" means that we set $\widehat{N}_{kt} = MB_k$ for each $1 \le k \le p$ and run the sorting procedure with these oracle sets. The results for "10 Most Correlated" use 20% of the data to specify $\widehat{N}_{kt}$ as the 10 most Pearson-correlated variables (in absolute value) to $X_k$ for each $1 \le k \le p$, and the other 80% of the data to estimate the topological ordering.

It is encouraging to see in Figure 5 that the accuracy of our method is high even for such a challenging high-dimensional setting. In fact, the average error rate is quite comparable to that for the smaller networks reported in Figure 2. As expected, an accurate neighborhood set provides better sorting results. Further, our method can run relatively quickly for large $p$, but its accuracy naturally is dictated by sample size. Figure 9 in Appendix B contains the sorting times to go along with Figure 5.

## 3.3 Application: Single-Cell Gene Expression Data

We now present an application of our method on the data of Yao et al. (2021)[1]. We seek a linear SEM to model a gene regulatory network, where each $X_k$ in Equation 1 is the expression level of a gene. We focused our attention on their dataset for which isolated single cells were processed for RNA sequencing using SMART-Seq v4 (labeled "Mouse Cortex+Hipppocampus (2019/2020)"). Noting Yao et al. (2021)'s finding that cells' gene expressions cluster according to region and cell type, the focus is a subset of the data

---

[1]Available at `http://cells.ucsc.edu/?ds=allen-celltypes+mouse-cortex&meta=regionlabel` in compressed TSV format.

as follows. We focused on glutamatergic cells from the mice brains' primary visual cortex. We also focus on cells for which injection materials are not specified (see Saleeba et al. (2019) for background on neuronal tracers). This takes us from 74,973 cells down to 7,159–the largest subset of all cell class, isocortex location, and injection material combinations. Moreover, a sizable amount of genes had expression measurements of exactly 0, so genes are also subset to those which were measured to be non-zero in 50% or more of these cells. This took us from 45,768 to 10,012 gene expression measurements taken across 7,159 cells.

### 3.3.1 Comparison to another scalable linear SEM estimation procedure

As for large simulated networks in Section 3.2, DirectLiNGAM and HighDimLiNGAM were too slow for this application. In order to compare ScoreLiNGAM to another linear structural equation modeling procedure, we applied the package `sparsebn` (Aragam et al., 2019) to our data. This method is a score-based method that maximizes a regularized Gaussian likelihood over the DAG space based on Aragam & Zhou (2015). To make comparisons across 50 repetitions computationally viable, we randomly select $p = 1000$ of the original 10,012 genes. For each repetition, we randomly sampled 2,000 cells: half of the cells ($n = 1000$) were designated to be in the training set, and the other half in test set; each data matrix was standardized such that columns have sample standard deviation 1 and sample mean 0.

In the training set, 20% of cells were randomly selected to estimate the Pearson correlation matrix. We specified the neighborhoods, $\widehat{N}_k$, for ScoreLiNGAM as the 50 genes $j \in [1000]\setminus\{k\}$ with the largest Pearson correlation (in absolute value) with gene $k$. The remaining 80% of training data was used to estimate a topological ordering and the linear SEM's coefficients (via ordinary least squares). For Sparsebn, no a priori neighborhood selection was used: parent sets for the linear SEM were learned with 100% of the training data using default options in the `estimate.dag` command, and the selection of the final DAG in the solution path was done by the recommended `select.parameter` command. For Sparsebn, the linear SEM's model parameters were estimated according to the selected DAG. Moreover, the noise densities fit to the residuals in the training set are either Gaussian or Laplace.

As can be seen in Figure 6, the Laplace density specification for the additive errors provides a significantly higher mean log-likelihood on the test set compared to a Gaussian density for both methods. This shows that the Laplace distribution, with its thicker tails than the Gaussian distribution, fits this data better. Furthermore, ScoreLiNGAM showed substantially higher test-data likelihood than Sparsebn under both error distributions for calculating the likelihood.

### 3.3.2 Application of ScoreLiNGAM to All 10,012 Genes

We now present the application of ScoreLiNGAM to all original $p = 10,012$ genes discussed at the start of this section. The application is as follows:

1. We randomly split the gene expression measurement matrix with 7,159 cells (rows) into two folds having 358 ($\sim 5\%$) and 6801 ($\sim 95\%$) of the cells, respectively.

2. On the first fold, we ran neighborhood linear regressions in which we restrict coefficients to be non-negative via `R`'s `glmnet` package (Friedman et al., 2010) with no ridge or lasso regularization. We then selected the sets $\widehat{N}_k$ to correspond to genes $\widehat{N}_k \subseteq [p]\setminus\{k\}$ such that coefficients are non-zero. The non-negative coefficient constraint for each linear regression, known as non-negative least squares, can itself be seen as a form of regularization that gives a sparse solution to the coefficient vectors (Slawski & Hein, 2013). Indeed, this constraint in the neighborhood regressions resulted in neighborhood sets of size 45 to 277 genes (of 10,011 possible genes), with a median of 145 genes per neighborhood set. Our use of non-negative least squares regression is motivated by the use of non-negative linear regression to impute single-cell gene expression measurements (Li & Li, 2018). It is also computationally faster compared to neighborhood Lasso regression.

3. We then randomly split the second fold into two folds, 2A and 2B, having 3401 ($\sim 47.5\%$) and 3400 ($\sim 47.5\%$) of the original cells, respectively.

4. On Fold 2A, we estimated the permutation $\hat{\pi}$ using ScoreLiNGAM.

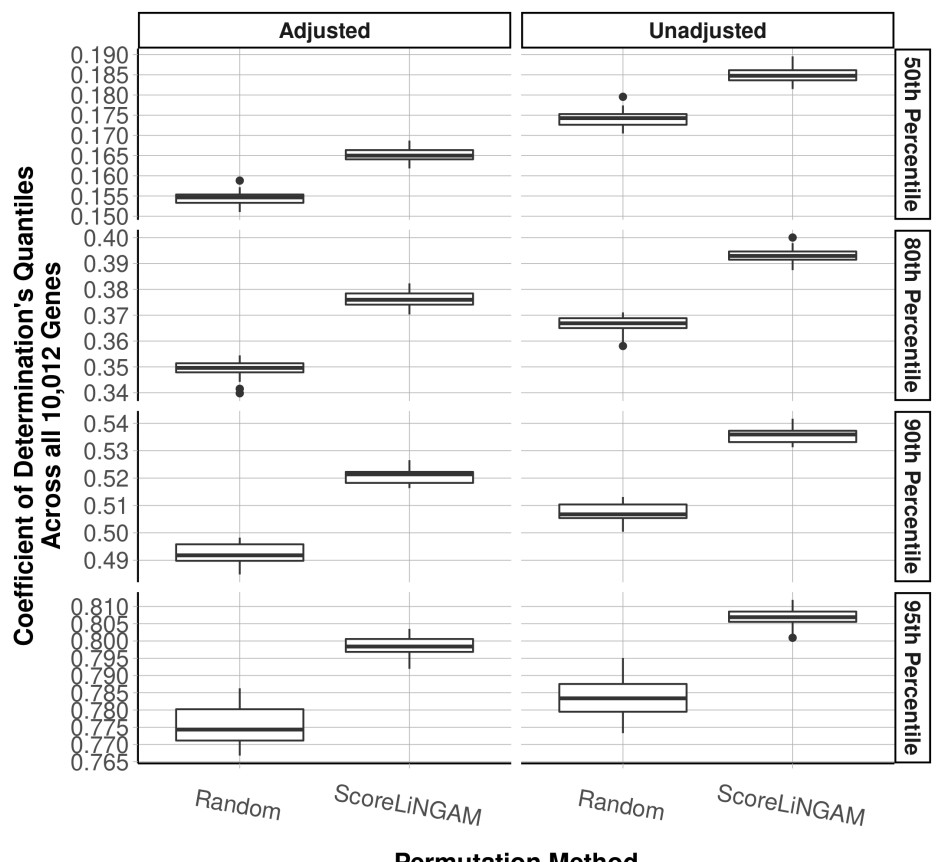

Figure 7: Across 30 replications, a comparison of the estimated coefficient of determination on Fold 2B for each gene. We summarize the coefficient of determination across genes by taking the median, 80th, 90th, and 95th percentiles.

5. On Fold 2B, the validation fold, we estimated via linear least squares regression the 10,012 by 10,012 weighted adjacency matrix for the DAG corresponding to the topological ordering defined by $\hat{\pi}$ and such that the support of the $k$-th column is the set of indices $\widehat{N}_k \cap \{\hat{\pi}(j)\}_{0 < j < \hat{\pi}^{-1}(k)}$.

6. Using the weighted adjacency matrix estimate from the previous step, we then calculated linear least squares residuals from the predictions given by the estimated parent set for each node $k$, on Fold 2B.

7. With the residuals of the last step, we calculated the coefficient of determination ($R_k^2$): for each gene $k \in [10,012]$, $R_k^2$ is an estimate of the proportion of variation explained by linearly regressing gene $k$'s measurement on the estimated parent genes. It is 1 minus the ratio of gene $k$'s residual sum of squares and the total sum of squares.

The result of this application is summarized in Figure 7. In order to make a comparison, we also calculated residuals from the linear SEM induced by a randomly generated topological ordering, denoted in Figure 7 as "Random." The two linear SEMs, with topological ordering given by ScoreLiNGAM or randomly generated, select parent sets as the intersection of a node's neighborhood set, $\widehat{N}_k$, and the node's predecessors in the corresponding topological ordering if any. We repeat Steps 3-7 a total of 30 times.

Considering that some genes may have more estimated parents than others, and that a coefficient of determination can be artificially large as the number of regressors increases, Figure 7 also includes the adjusted coefficient of determination which incorporates a penalty for the number of regressors (Neter et al.,

1983). The adjusted coefficient of determination is

$$1 - (1 - R_k^2)\frac{n_B - 1}{n_B - |\widehat{PA_k}|},$$

where $|\widehat{PA_k}|$ is the number of estimated parents for node $k$ and $n_B = 3400$ is the sample size in Fold 2B.

As we can see from Figure 7, ScoreLiNGAM gives higher coefficients of determination (adjusted and unadjusted) on the test datasets (validation fold 2B) compared to a randomly generated permutation across all random replications–as summarized by the median, 80th, 90th, and 95th percentiles taken across the 10,012 coefficients of determination. Taking for granted the linearity assumption, the above higher-end percentiles of the $R_k^2$ across all genes provide meaningful comparisons because it may very well be the case that a majority of the genes have quite random expression patterns. Based on the 90th percentile for the adjusted coefficient of determination in Figure 7, it appears that for 10% of all the genes, more than 51% of their expression variation is explainable by its estimated parents in the linear SEM given by ScoreLiNGAM. As shown by the 95th percentiles in Figure 7, for the top 5% of genes in terms of adjusted $R^2$, 79% or more of the genes' expression variation is explainable by its estimated parents in the linear SEM given by ScoreLiNGAM. These $R^2$ levels are significantly higher than those given by random permutations, with no overlap in the boxplots in Figure 7.

Across the 30 replications, ScoreLiNGAM's sorting time for all 10,012 genes had a median of 10.28 minutes, confirming its scalability for such large and high-dimensional datasets of $p > 10,000$ and $n > 3,000$.

## 4    Discussion

In this paper, we claimed that sequentially applying Algorithm 1 can give interesting structure learning results. We demonstrated this with a novel sequential procedure based on parametric specification that provides an alternative to the state of the art for the identifiability and estimation of a linear DAG model with non-Gaussian errors. We discussed the conditions, Assumptions 2.1 and 2.2, under which the proposed causal discovery procedure will identify the valid DAG. We also proposed a relatively simple procedure that can make strategic use of an a priori known neighborhood set for each node. Finally, we presented numerical evidence that our procedure scales to large dimensions, which is otherwise not the case for the state-of-the-art of LiNGAM. We accompanied these simulations with a real-data application.

Our motivation for learning a topological ordering is that it is a relatively simpler problem to tackle compared to searching across the super-exponentially large space of DAG structures. Under Assumption 2.3 and regardless of whether the order estimate satisfies Definition 1.1, a graph output by our Algorithm 2 will have false positive edges corresponding to at least the number of times two non-adjacent nodes have a common child in the true graph. This is because a Markov blanket contains all co-parents (or "spouses"). Moreover, the number of incorrectly oriented true edges will be the total number of true edges times the accuracy metric in Equation equation 11. Thus, to produce a good graph estimate given our estimated ordering, one needs an additional step to apply a sparse regression of each $X_j$ on the variables sorted in front of it to estimate the support (parent set). This is the well-studied support recovery problem in sparse linear regression.

With regard to the linearity, scale-location family, and a priori known Markov blanket assumptions for Theorem 2.5, we concede to the apparent limitation compared to assuming non-linear relations and making less parametric assumptions. However, to us, there appears to be a trade-off between more general modeling assumptions and scalability to a large number of variables, the latter of which our paper addresses. Recall the application to the large-scale gene expression data in § 3.3.2, which demonstrates our method's improvement at linearly explaining variation in a large number of genes compared to a random permutation baseline. The concern about whether the entries of the error vector need to follow the same distribution family is an important one. It is not immediately clear how our current theoretical results, making use of information theoretic quantities, will come about without this assumption. Extensions of our work may like to explore whether this assumption can be relaxed. This relaxation is suggested by our simulation results where the errors' distribution family is mis-specified.

As a practical manner, consider prospective applications to single cell gene expression data (scRNA-seq) as in § 3.3. Recent work suggests a hierarchical structure between true (hidden) expressions and measured expressions with missing, and possibly zero, counts (Sarkar & Stephens, 2021). Should the hierarchical nature be justified, further work on causal discovery for gene co-expression models may need to incorporate the fact that what we really would like is a graphical model, possibly causal if a domain expert agrees, on the true (hidden) expressions. Along these lines, future causal discovery procedures for such data can build on the procedures of McDavid et al. (2019) and Yu et al. (2020), which themselves build on Gaussian graphical models, using a heavier tail distribution for residuals as we do here.

Further extensions of the work presented here include formal statistical guarantees along with extensions of the likelihood ratio approach to nonlinear SEMs.

## 5 Broader Impact Statement

Our scalable causal discovery method comes with extra modeling assumptions, not to mention the fact that we are making use of the causal sufficiency assumption (no unobserved confounders or selection bias). Therefore, it is important for a responsible practitioner to not over-extend their conclusions with this or related data mining methods.

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

# A    Proof of Theorem 2.5

## A.1    Formal Proof of Theorem 2.5

*Proof of Theorem 2.5.*
Our proof boils down to making the correct decision in Algorithm 1 at step 1, then making the correct choice at step 2 assuming the choice in step 1 was correct, and so on.

For the sake of induction, let us assume that $\mathcal{A}_t$ is correct in the sense that $PA_a \subseteq \mathcal{A}_t$ for all $a \in \mathcal{A}_t$. This is true at the base case $t = 1$ when $\mathcal{A}_t = \emptyset$, since having made no ordering choices also means we have made no mistakes.

Let $k \in S_t$ be an invalid node to continue the ordering in the sense that $PA_k \cap \mathcal{A}_t \neq \emptyset$. And let $\ell \in S_t$ be a valid node to continue the ordering in the sense that $PA(\ell) \subseteq \mathcal{A}_t$.

Lemma 2.7 tells us that the least squares residual $R_{\ell t} \sim f_{\ell t}(r_{\ell t})$ is no closer to Gaussian than $R_{kt} \sim f_{kt}(r_{kt})$ in the sense that:

$$D_{KL}\left(f_{kt}(r_{kt})||\phi(r_{kt};\sigma_{kt})\right) \leq D_{KL}\left(f_{\ell t}(r_{\ell t})||\phi_{\ell t}(r_{\ell t})\right)$$

Furthermore, regularity Assumption 2.2 ensures that:

$$D_{KL}\left(f_{kt}(r_{kt})||g_k(r_{kt};\eta_{kt})\right) > 0.$$

On the other hand, so long as we properly specified the error density for node $\ell$, we have that:

$$D_{KL}\left(f_{\ell t}(r_{\ell t})||g_\ell(r_{\ell t};\eta_{\ell t})\right) = 0.$$

Thus,

$$
\begin{aligned}
\mathbb{E}_{f_{kt}(r_{kt})}\left[\log \frac{g_k(R_{kt};\eta_{kt})}{\phi(r_{kt};\sigma_{kt})}\right] &= D_{KL}\left(f_{kt}(r_{kt})||\phi(r_{kt};\sigma_{kt})\right) - D_{KL}\left(f_{kt}(r_{kt})||g_k(r_{kt};\eta_{kt})\right) \\
< \mathbb{E}_{f_{\ell t}(r_{\ell t})}\left[\log \frac{g_\ell(R_{\ell t};\eta_{\ell t})}{\phi_{\ell t}(R_{\ell t})}\right] &= D_{KL}\left(f_{\ell t}(r_{\ell t})||\phi_{\ell t}(r_{\ell t})\right).
\end{aligned}
$$

Altogether, this implies that

$$\max_{j \in S_t} \mathcal{S}(j, \mathcal{A}_t) > \mathcal{S}(k, \mathcal{A}_t).$$

and

$$\ell = \arg\max_{j \in S_t} \mathcal{S}(j, \mathcal{A}_t),$$

since $\ell$ and $k$ were arbitrary valid and invalid nodes, respectively.

So at step $t$, we will always make the correct choice for a node to continue the ordering.    □

## A.2    Proofs of Lemma 2.6 and Lemma 2.7

In this section, we formally prove Lemma 2.6 and Lemma 2.7.

### A.2.1    Some Useful Shorthand Notation

Let us define some new strategic sets which contain indices in $[p]$, and review some we have been using already.

- The set

$$
\mathcal{A}_t = \begin{cases} \emptyset & t = 1 \\ \{\hat{\pi}(1), \ldots, \hat{\pi}(t-1)\} & t \geq 2 \end{cases}.
$$

  This is the partial ordering at step $t = 1, 2, \ldots$. In our population-level identification results, we will typically assume it is correct at step $t$, which means that for all $a \in \mathcal{A}_t$, $PA_a \subset \mathcal{A}_t$.

- $S_t = [p] \backslash \mathcal{A}_t$ is the set of unordered nodes at step $t$.

- $MB_k = PA_k \cup CH_k \cup_{j \in CH_k} PA_j$ is the Markov Blanket of node $k$.

- $\widehat{N}_k$ is the Markov Blanket superset such that $\widehat{N}_k \supseteq MB_k$. In finite data, we will typically estimate $\widehat{N}_k$ by a procedure such as neighborhood lasso regression, so this containment may not hold. For the sake of this section, because we are deriving quantities at the population-level, we assume that $\widehat{N}_k$ is known and contains the true Markov blanket. Note that trivially, we may consider $\widehat{N}_k = [p] \backslash \{k\}$, and the results of this section would still hold.

- $\widehat{N}_{kt} = \mathcal{A}_t \cap \widehat{N}_k$ is the intersection of the Markov blanket superset with the partial ordering.

- $L_{kt} = \bigcup_{j \in \widehat{N}_{kt}} \{j\} \cup AN_j$, which are either nodes of $\widehat{N}_{kt}$ or ancestors of nodes in $\widehat{N}_{kt}$. When $\mathcal{A}_t$ is correct, it is necessarily the case that $L_{kt} \subseteq \mathcal{A}_t$ for each $k \notin \mathcal{A}_t$.

- $L_{kt}^C$, the complement of set $L_{kt}$ which either contains nodes in $\mathcal{A}_t$ which are not in $L_{kt}$, i.e. the nodes of $\mathcal{A}_t \backslash L_{kt}$, or which are unordered, i.e. we have that $S_t \subseteq L_{kt}^C$.

Note that for each node $k \in S_t$ we can write:

$$
X_k = \mathbf{M}_{k\cdot}\epsilon = \mathbf{M}_{kL_{kt}}\epsilon_{L_{kt}} + \mathbf{M}_{kL_{kt}^C}\epsilon_{L_{kt}^C}, \tag{12}
$$

where the second equality holds since $L_{kt} \cup L_{kt}^C = [p]$. We can similarly write

$$
X_{\widehat{N}_{kt}} = \mathbf{M}_{\widehat{N}_{kt}\cdot}\epsilon = \mathbf{M}_{\widehat{N}_{kt}L_{kt}}\epsilon_{L_{kt}}. \tag{13}
$$

We omit a term with $\epsilon_{L_{kt}^C}$ since by definition of $L_{kt}$, the sub-mixing matrix $\mathbf{M}_{\widehat{N}_{kt}L_{kt}^C}$ is a zero matrix.

Combining Equation 12 and Equation 13,

$$
R_{kt} = \left( \mathbf{M}_{kL_{kt}} - \beta_{kt}^T \mathbf{M}_{\widehat{N}_{kt}L_{kt}} \right) \epsilon_{L_{kt}} + \mathbf{M}_{kL_{kt}^C}\epsilon_{L_{kt}^C},
$$

which we will make use of in the proof for Lemma 2.6 below.

### A.2.2 Proof of Lemma 2.6

*Proof of Lemma 2.6.*
**Case 1:** Assume $k$ is a valid node to continue the ordering in the sense that $PA_k \subseteq \mathcal{A}_t$. We want to show that $R_{kt} = \epsilon_k$. In this case, $\mathbf{M}_{kL_{kt}^C}$ has a non-zero entry corresponding to only $\mathbf{M}_{kk} = 1$. This is because $AN_k = \text{support}(\mathbf{M}_{k\cdot}) \backslash \{k\} \subseteq L_{kt}$, which in turn holds because $PA_k \subseteq MB_k \cap \mathcal{A}_t \subseteq \widehat{N}_{kt}$. Thus we have

$$
\mathbf{M}_{kL_{kt}^C}\epsilon_{L_{kt}^C} = \epsilon_k.
$$

So we have left to show that

$$
\left( \mathbf{M}_{kL_{kt}} - \beta_{kt}^T \mathbf{M}_{\widehat{N}_{kt}L_{kt}} \right) \epsilon_{L_{kt}} = 0.
$$

Recall that $\mathbf{B}$ is the weighted adjacency matrix for the underlying LiNGAM. We have that $\text{support}(\mathbf{B}_{\cdot k}) = PA_k$. Let us index the entries of the column vector $\mathbf{B}_{\cdot k}$ by $\widehat{N}_{kt}$ and denote this as $\mathbf{B}_{\widehat{N}_{kt}k}$. One thing that could be helpful to prove is that if $k$ is valid, then:

$$
\beta_{kt} = \mathbf{B}_{\widehat{N}_{kt}k}.
$$

Because support$(\mathbf{B}_{\cdot k}) = PA_k$ and $PA_k \subseteq \widehat{N}_{kt}$, consider that

$$f X_k = X^T \mathbf{B}_{\cdot k} + \epsilon_k = X_{\widehat{N}_{kt}}^T \mathbf{B}_{\widehat{N}_{kt}k} + \epsilon_k,$$

with $\epsilon_k \perp\!\!\!\perp X_{\widehat{N}_{kt}k}$ and $\mathbb{E}[\epsilon_k] = 0$. Thus,

$$
\begin{aligned}
\beta_{kt} &= \left( \mathbb{E}\left[ X_{\widehat{N}_{kt}} X_{\widehat{N}_{kt}}^T \right] \right)^{-1} \left( \mathbb{E}\left[ X_{\widehat{N}_{kt}} X_{\widehat{N}_{kt}}^T \right] \mathbf{B}_{\widehat{N}_{kt}k} + \mathbb{E}\left[ X_{\widehat{N}_{kt}} \epsilon_k \right] \right) \\
&= \left( \mathbb{E}\left[ X_{\widehat{N}_{kt}} X_{\widehat{N}_{kt}}^T \right] \right)^{-1} \mathbb{E}\left[ X_{\widehat{N}_{kt}} X_{\widehat{N}_{kt}}^T \right] \mathbf{B}_{\widehat{N}_{kt}k} \\
&= \mathbf{B}_{\widehat{N}_{kt}k},
\end{aligned}
\tag{14}
$$

as we wanted.

It follows that $X_k = \mathbf{B}_{\widehat{N}_{kt}k}^T X_{\widehat{N}_{kt}} + \epsilon_k = \beta_{kt}^T X_{\widehat{N}_{kt}} + \epsilon_k$. This then means that $R_{kt} = X_k - \beta_{kt}^T X_{\widehat{N}_{kt}} = \epsilon_k$, as we wanted to show.

**Case 2:** Assume $k$ is not a valid node. All we need in this case for our identifiability proof is that $R_{kt}$ is a linear combination of more than one independent component. This is the case because if $k$ is invalid to continue the ordering, then we have that there exists at least one $j \in PA_k$ such that $j \in S_t$ (unordered) and therefore $j \in L_{kt}^C$. Recall that:

$$R_{kt} = \left( \mathbf{M}_{kL_{kt}} - \beta_{kt}^T \mathbf{M}_{\widehat{N}_{kt}L_{kt}} \right) \epsilon_{L_{kt}} + \mathbf{M}_{kL_{kt}^C} \epsilon_{L_{kt}^C}.$$

Note that it is necessarily the case that $\mathbf{M}_{kj} \neq 0$, otherwise $j \notin PA_k$. Thus, $R_{kt}$ includes the sum $\mathbf{M}_{kj}\epsilon_j + \epsilon_k$. That is, $R_{kt}$ in this case is a linear combination of more than one independent component in $\epsilon$. Note that $R_{kt}$ could be a linear combination of more entries in $\epsilon$, in addition to $\epsilon_j$ and $\epsilon_k$.

$\square$

### A.2.3 Some Information Theory Definitions and Results

We now present some straightforward information theoretic results. They are meant to help demonstrate that our surrogate optimization (now a likelihood ratio) approach for Algorithm 1 leads to the identifiability of a causal order. These lemmas are used later to prove Lemma 2.7, a key result that says valid nodes $j$ in a LiNGAM are no closer to Gaussian compared to invalid nodes $k$, conditional on the nodes in $\widehat{N}_{jt}$ and $\widehat{N}_{kt}$, respectively.

**Definition A.1** (Differential Entropy).
For a continuous random variable $X$ with density $p(x)$, denote $\mathbb{E}_{p(x)}[\cdot]$ to be expectation with respect to $p(x)$. The differential entropy of $X$ is given by:

$$\mathbf{h}(X) = \mathbb{E}_{p(x)}\left[ \log \frac{1}{p(X)} \right].$$

**Lemma A.2** (Restatement of the entropy power inequality).
*Consider two independent random variables $X \sim p(x)$ and $Y \sim p(y)$, and let $X' \sim \mathcal{N}\left(\mathbb{E}[X'], \mathbb{V}[X']\right)$ and $Y' \sim \mathcal{N}\left(\mathbb{E}[Y'], \mathbb{V}[Y']\right)$ be independent random variables such that $\mathbf{h}(X) = \mathbf{h}(X')$ and $\mathbf{h}(Y) = \mathbf{h}(Y')$. Then:*

$$\mathbf{h}(X + Y) \geq \mathbf{h}(X' + Y').$$

*Proof.* This is exactly *Theorem 17.8.1* of Cover & Thomas (2005), so we refer the reader to their proof.

$\square$

**Lemma A.3** (KL Divergence from Gaussianity).
*Let $X \sim p(x)$ and $q(x)$ the density for $\tilde{X} \sim \mathcal{N}\left(\mathbb{E}[X], Cov[X]\right)$.*

$$D_{KL}\left(p(x)||q(x)\right) = \mathbf{h}(\tilde{X}) - \mathbf{h}(X). \tag{15}$$

*As in Lemma A.2, let $X' \sim \mathcal{N}\left(\mathbb{E}[X'], \mathbb{V}[X']\right)$ such that $\mathbf{h}(X) = \mathbf{h}(X')$. We can equivalently write the KL divergence from Gaussianity as:*

$$D_{KL}\left(p(x)||q(x)\right) = \frac{1}{2}\log\left(\frac{\mathbb{V}[X]}{\mathbb{V}[X']}\right).$$

*Proof.*
Because

$$\mathbf{h}(\tilde{X}) = \mathbb{E}_{\tilde{X}\sim q(x)}\left\{\log\frac{1}{q(\tilde{X})}\right\} = \mathbb{E}_{X\sim p(x)}\left\{\log\frac{1}{q(X)}\right\},$$

by properties of this normal distribution (namely, that $\mathbb{E}[\log q(X)] \propto \mathbb{V}[X] = \mathbb{V}[\tilde{X}]$) we have that:

$$D_{KL}\left(p(x)||q(x)\right) = \mathbf{h}(\tilde{X}) - \mathbf{h}(X).$$

Noting that the differential entropy for any $\mathcal{N}(\mu, \sigma^2)$ is $\frac{1}{2}\log(2\pi e\sigma^2)$ and our assumption that $\mathbf{h}(X) = \mathbf{h}(X')$, we arrive at the second equality:

$$D_{KL}\left(p(x)||q(x)\right) = \frac{1}{2}\log\left(\frac{2\pi e\mathbb{V}[X]}{2\pi e\mathbb{V}[X']}\right).$$

Note that also $\mathbb{V}[X] = \mathbb{V}[\tilde{X}] \geq \mathbb{V}[X'] \iff D_{KL}\left(p(x)||q(x)\right) \geq 0$, which is the case because KL-divergence is always non-negative.

$\square$

This well known result also implies that the normal distribution is the maximum entropy distribution when we constrain the first and second order moments of each distribution to be the same.

**Lemma A.4** (Same distance to Gaussianity).
*Let $\tilde{\epsilon}_k \sim \mathcal{N}(0, \mathbb{V}[\epsilon_k])$ with density $q_k(\cdot)$ for each $k = 1, 2, \ldots, p$. Also let $\epsilon'_k \sim \mathcal{N}(0, \mathbb{V}[\epsilon'_k])$ such that $\mathbf{h}(\epsilon'_k) = \mathbf{h}(\epsilon_k)$. If $\epsilon$ in our LiNGAM satisfies Assumption 2.1, then there exists a constant $\gamma \geq 0$ such that*

$$D_{KL}\left(g(\epsilon_k; \theta_k)||q_k(\epsilon_k)\right) = \gamma$$

*and*

$$\frac{\mathbb{V}[\epsilon_k]}{\mathbb{V}[\epsilon'_k]} = \tilde{\gamma} = \exp(2\gamma)$$

*for all $k = 1, 2, \ldots, p$.*

*Proof.*
From Lemma A.3, we have that:

$$D_{KL}\left(g(\epsilon_k; \theta_k)||q_k(\epsilon_k)\right) = \mathbf{h}(\tilde{\epsilon}_k) - \mathbf{h}(\epsilon'_k).$$

Noting Assumption 2.1 and properties of differential entropy under a rescaling, it follows that for $U \sim g(\cdot; \theta_0)$:

$$\mathbf{h}(\epsilon_k) = \mathbf{h}(U) + \log(\theta_k/\theta_0).$$

Let $U' \sim \mathcal{N}(0, \mathbb{V}[U'])$ such that $\mathbf{h}(U') = \mathbf{h}(U)$. We have also that

$$\mathbf{h}(\epsilon'_k) = \mathbf{h}(U') + \log(\theta_k/\theta_0),$$

based on the construction of both $\epsilon_k'$ and $U'$.

Similar to $\tilde{\epsilon}_k$, let $U \sim \mathcal{N}(0, \mathbb{V}[U])$. Thus, regardless of $k = 1, 2, \ldots, p$, we have that:

$$\frac{1}{2} \log \left( \frac{2\pi e \mathbb{V}[\epsilon_k]}{2\pi e \mathbb{V}[\epsilon_k']} \right) = \mathbf{h}(\tilde{\epsilon}_k) - \mathbf{h}(\epsilon_k') = \mathbf{h}(U) - \mathbf{h}(U') =: \gamma.$$

$\square$

### A.2.4 Proof of Lemma 2.7

*Proof of Lemma 2.7.*
For each $j \in [p]$, let $\epsilon_j'$ be a normally distributed random variable such that $\mathbf{h}(\epsilon_j') = \mathbf{h}(\epsilon_j)$, while $\tilde{\epsilon}_j$ is distributed as $\mathcal{N}(\mathbb{E}[\epsilon_j], \mathbb{V}[\epsilon_j])$. Here, for all $j, k \in \{1, 2, \ldots, p\}$, $\tilde{\epsilon}_j \perp\!\!\!\perp \tilde{\epsilon}_k$ (unless $j = k$) and $\tilde{\epsilon}_j \perp\!\!\!\perp \epsilon_k'$ (even if $j = k$).

Recall also that for $j \in S_t$

$$R_{jt} = \left( \mathbf{M}_{jL_{kt}} - \beta_{jt}^T \mathbf{M}_{\widehat{N}_{jt}L_{tj}} \right) \epsilon_{L_{tj}} + \mathbf{M}_{jL_{tj}^C} \epsilon_{L_{tj}^C} = \sum_{i \in [p]} \delta_{ij} \epsilon_i,$$

where the coefficients $\delta_{ij}$ in the last equality are used for shorthand. Note that $\delta_{jj} = 1$ always. And if $j$ is invalid to continue the ordering, then also $\delta_{ij} \neq 0$ for at least one other $i \in [p] \backslash \{j\}$, based on Lemma 2.6.

The relation between the quantities of interest is as follows:

$$
\begin{aligned}
D_{KL}\left(f_{kt}(r_{kt}) \| \phi(r_{kt}; \sigma_{kt})\right) \quad &= \mathbf{h}(\tilde{R}_{kt}) - \mathbf{h}\left(R_{kt}\right) && \text{by Lemma A.3} \\
&= \mathbf{h}\left( \sum_{i \in [p]} \delta_{ik} \tilde{\epsilon}_i \right) - \mathbf{h}\left( \sum_{i \in [p]} \delta_{ik} \epsilon_i \right) && \text{Notice: } \tilde{R}_{kt} \overset{d}{=} \sum_{i \in [p]} \delta_{ik} \tilde{\epsilon}_i \\
&\leq \mathbf{h}\left( \sum_{i \in [p]} \delta_{ik} \tilde{\epsilon}_i \right) - \mathbf{h}\left( \sum_{i \in [p]} \delta_{ik} \epsilon_i' \right) && \text{by Lemma A.2} \\
&= \frac{1}{2} \log \left( \frac{2\pi e \sum_{i \in [p]} \delta_{ik}^2 \mathrm{Var}(\epsilon_i)}{2\pi e \sum_{i \in [p]} \delta_{ik}^2 \mathrm{Var}(\epsilon_i')} \right) && \text{by normality of the } \tilde{\epsilon}_i, \epsilon_i' \\
&= \frac{1}{2} \log \left( \frac{2\pi e \tilde{\gamma} \sum_{i \in [p]} \delta_{ik}^2 \mathrm{Var}(\epsilon_i')}{2\pi e \sum_{i \in [p]} \delta_{ik}^2 \mathrm{Var}(\epsilon_i')} \right) && \text{by Lemma A.4} \\
&= \gamma \\
&= D_{KL}\left(f_{\ell t}(r_{\ell t}) \| \phi_{\ell t}(r_{\ell t})\right),
\end{aligned}
$$

(16)

as we wanted (Recall $R_{\ell t} = \epsilon_\ell$ by Lemma 2.6).

$\square$

## B More Figures

### B.1 Sorting Time for Small Networks

The takeaway of Figure 8 is that ScoreLiNGAM is generally much faster. Consider the largest DAG, the Andes network ($p = 223$), where the sorting time of ScoreLiNGAM is typically under 1 second across all sample sizes, while for HighDimLiNGAM (parallelized across 7 threads) the sorting procedure takes between 10-1000 seconds across sample sizes. We note that ScoreLiNGAM is written with `C++` using the `Armadillo` linear algebra library and an `R` wrapper via the `Rcpp` package, while DirectLiNGAM is written in `Python` (https://github.com/cdt15/lingam) with a wrapper function in `R` using the `reticulate` package that is written by this paper's authors. HighDimLiNGAM is also written in `C++` (https://github.com/ysamwang/highDNG)

with an `R` wrapper, but it searches regressor subsets when computing low-dimensional linear regressions–the likely reason for its slower time despite 7 parallel threads. All simulations were run on a Dell XPS 13 with Intel Core™ i7-8550U CPU @ 1.80GHz × 8, 8 GB RAM, and 64-bit Ubuntu 20.04.3 LTS OS.

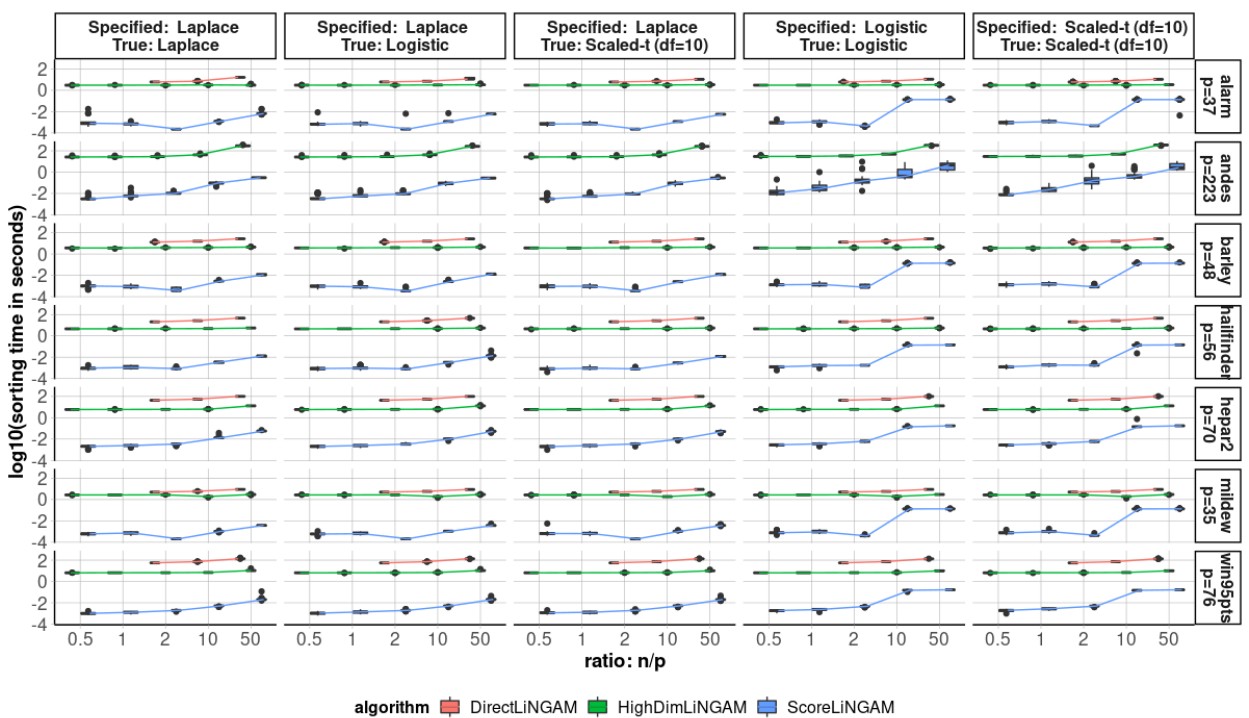

Figure 8: The simulation times for LiNGAM estimation procedures

## B.2 Sorting Times for Large Networks

Figure 9 contains the sorting times to go along with Figure 5 in the main text.

## B.3 Further simulations: Results without re-scaling the data matrix

See Figure 10 for results on the alarm, barley, and mildew networks.

## B.4 Further simulations: Results without re-scaling the data matrix and neighborhood estimation

See Figure 11 for results on the barley, alarm, and mildew networks.

# C  Obtaining the scale-parameter for Empirical Mean Log-likelihood in Equation 9

As discussed in the main text, our sequential algorithm at step $t \geq 1$ in practice requires the estimation of the scale parameter, $\eta_{kt}$, in Equation 9. Here, we discuss the estimator for the three parametric assumptions used in this paper. We make use of the respective definitions and properties in Forbes et al. (2010).

- **Laplace Distribution:** If $\epsilon_k \sim \text{Laplace}(0, \theta_k)$, we have that $\theta_k = \mathbb{E}[|\epsilon_k|]$ is the scale parameter. When $g(\cdot; \eta_{kt})$ is specified as the density for $\text{Laplace}(0, \eta_{kt})$, the maximum likelihood estimator we use in practice is $\hat{\eta}_{kt} = \frac{1}{n} \left\| \hat{R}_{kt} \right\|_1$.

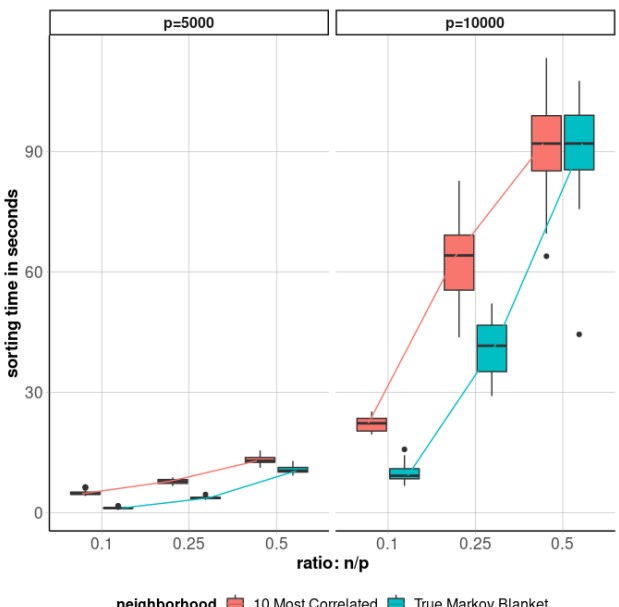

Figure 9: Sorting times for ScoreLiNGAM under $p = 5000, 10000$ and $n = 0.1p, 0.25p, 0.5p$. Color indicates how the neighborhood sets are constructed.

- **Logistic Distribution:** If $\epsilon_k \sim \text{Logistic}(0, \theta_k)$, then $\theta_k$ is the scale parameter. We have that $\text{Var}[\epsilon_k] = \frac{\pi^2}{3}\theta_k^2$. When $g(\cdot; \eta_{kt})$ is specified as the density for $\text{Logistic}(0, \eta_{kt})$, we find that the plug-in estimator $\hat{\eta}_{kt} = \frac{\sqrt{3}}{\pi}\hat{\sigma}_{kt}$ to work satisfactorily.

- **Scaled-t Distribution:** If $\epsilon_k \sim \text{Scaled-t}(0, \nu, \theta_k)$, then we say $\epsilon_k$ is equal in distribution to the scale parameter, $\theta_k$, times $U \sim t(0, \nu)$, a Student's t-distributed random variable having mean 0 and degrees of freedom $\nu > 0$. That is, $\epsilon_k \overset{d}{=} \theta_k U$. For $\nu > 2$, we have that $\text{Var}[\epsilon_k] = \theta_k^2\left(\frac{\nu}{\nu-2}\right)$. When $g(\cdot; \eta_{kt})$ is specified as the density for $\text{Scaled-t}(0, \nu, \eta_{kt})$ with $\nu > 2$ assumed to be known, we find that the plug-in estimator $\hat{\eta}_{kt} = \hat{\sigma}_{kt}\sqrt{\frac{\nu-2}{\nu}}$ to work satisfactorily.

In Equation 9 and in the plug-in estimators for the Logistic and Scaled-t specifications, we use

$$\hat{\sigma}_{kt}^2 = \frac{1}{n}\left\|\hat{R}_{kt}\right\|_2^2.$$

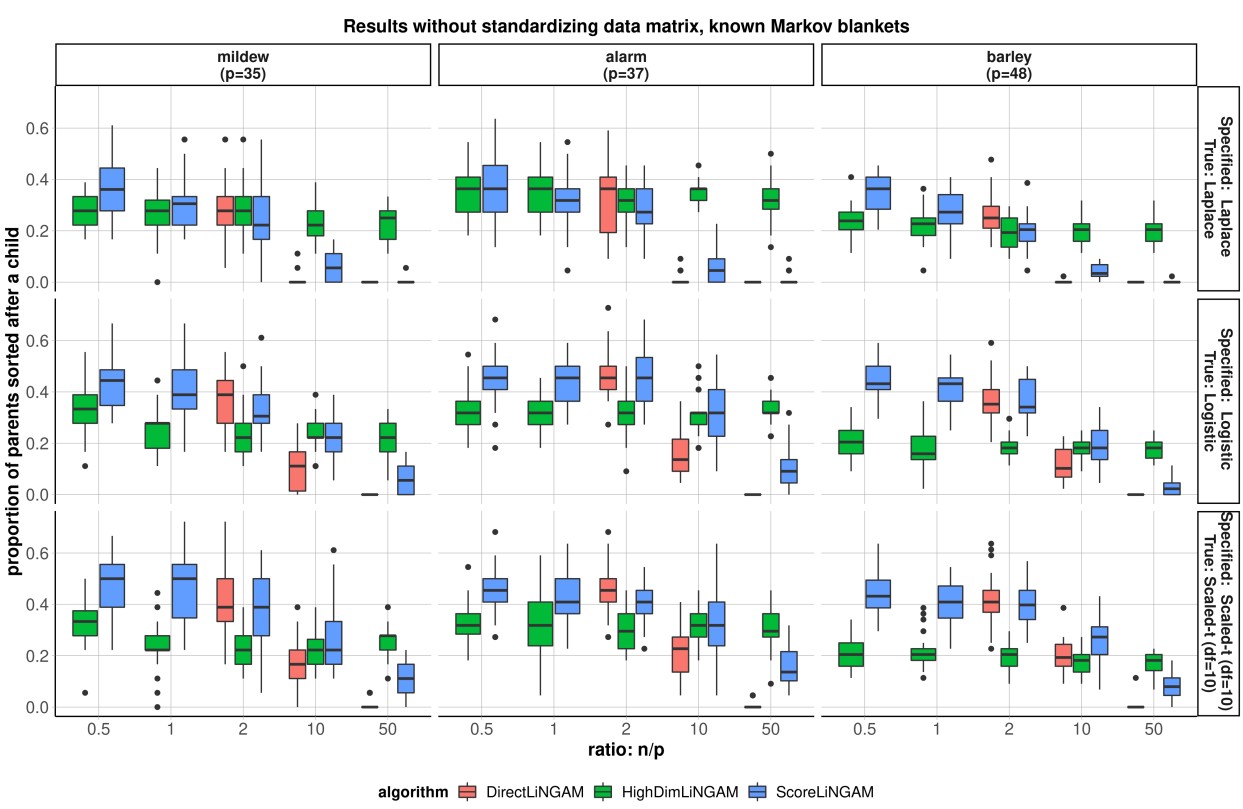

Figure 10: An extension of the simulation results in Figure 2 comparing LiNGAM estimation procedures. Here, we do not re-scale the data matrix.

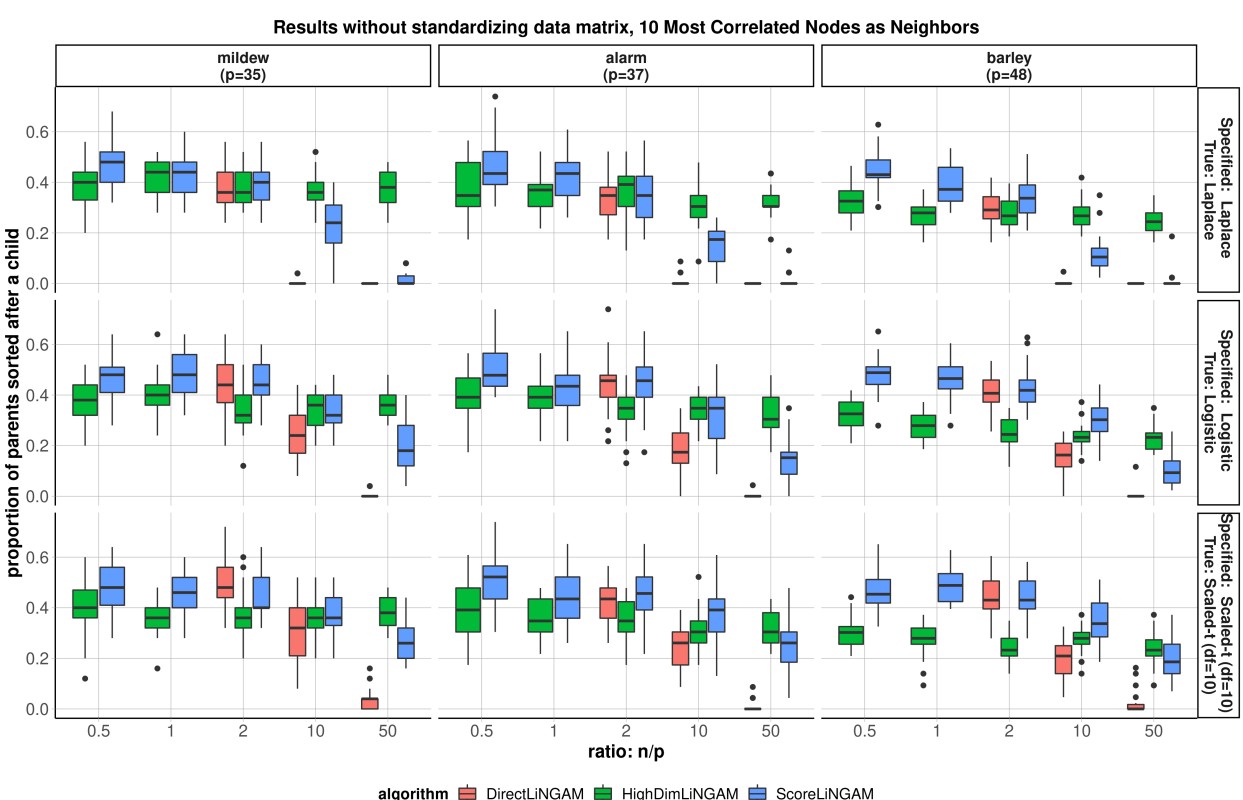

Figure 11: An extension of the simulation results in Figure 2 comparing LiNGAM estimation procedures. Here, we do not re-scale the data matrix. For ScoreLiNGAM and HighDimLiNGAM, we use 20% of the observations to estimate the correlation matrix and neighborhood sets, and the remaining 80% of observations to estimate the topological orderings. For DirectLiNGAM, all 100% of observations are used to estimate the ordering without neighborhood pre-specification

