# OpenReview forum: "Sequentially learning the topological ordering of directed acyclic graphs with likelihood ratio scores"
_TMLR — Accepted by TMLR_

### Review · Reviewer_5PmC · 2022-08-14

**Summary Of Contributions:**

The paper proposes a new approach for learning a topological node ordering of a DAG (causal ordering) when causal relations are linear with non-Gaussian noise, e.g. as in the first step of the LiNGAM algorithm, based on likelihood ratios of linear regression residuals.

The approach is $O(p^2)$ in general and $O(pd)$ when neighborhoods of each node are known or Markov blankets can be estimated accurately and $d$ is the largest neighborhood size.

The procedure is proven to be correct when noise terms belong to a non-Gaussian scale-location family density.

Experiments show the accuracy of the approach is favorable and more computationally efficient for smaller networks and it can scale to larger networks than can be handled by existing comparable approaches.

**Broader Impact Concerns:**

No concerns

**Requested Changes:**

Most important requested changes:

1) The paper would be greatly improved by providing a thorough discussion of the theoretical complexities of related methods for estimating causal orderings (and maybe adding a table listing each)

2) The paragraph preceding Theorem 2.5 does not provide an intuitive explanation for why selecting the most non-Gaussian ratio is correct. This becomes clear in the following section, but it would be helpful to explain the general idea at this stage in the paper.


Other possible changes that may improve the paper:

1) Better understanding the robustness of the algorithm, compared to existing approaches. when estimated Markov blanket sets are slightly incorrect would improve the paper

2) It's not clear whether the 'order estimation error' defined in (11) is new or has been used in existing approaches. It might improve the paper to also include a more well known ranking accuracy metric like Spearman correlation or at least provide more discussion for the justification of the proposed metric. Specifically, the proposed metric may overpenalize errors made at the beginning of the procedure when the ranking is otherwise correct compared to Spearman or a related metric.

**Strengths And Weaknesses:**

Strengths:

1) The paper is generally well written and technical matter is self contained

2) The proposed method allows existing LiNGAM-style methods to scale to larger variable sets when neighborhoods are known or Markov blankets can be estimated

3) The correctness results are rigorous

4) Simulations are comprehensive and results indicate a slight gain in accuracy


Weaknesses:

1) While the main contribution of the paper is a causal order estimation procedure that is more computationally efficient than existing approaches, the theoretical complexities of these approaches are not provided

2) The list of assumptions required for the procedure to work is somewhat extensive: linearity, non-Gaussianity, scale-location families, knowledge of neighborhoods

3) In general, the significance of the neighborhood assumption is somewhat glossed over - it's not clear how easy it is to estimate Markov blankets accurately in general and how much errors made during this estimation affect the procedure

4) The intuitive explanation of Theorem 2.5 could be better explained

---

> ### Author Response · Authors · 2022-09-30
> **Review of Paper313 by Reviewer 5PmC**
>
> We would like to thank the reviewer for taking the time to suggest edits that will strengthen the readability of this paper.
>
> - In the updated Section 1.2, we now contrast computational complexity of our Algorithm 2 and the other LiNGAM learning methods.
>
> - At the stage in the paper where Theorem 2.5 is stated, we now give better intuition rather than only doing so later in the paper.
>
>
> - With regard to the linearity, scale-location family, and a priori known Markov blanket assumptions, we agree at the apparent limitation compared to assuming non-linear relations and making less parametric assumptions. However, to us, there appears to be a trade-off between more general modeling assumptions and scalability to a large number of variables, the latter of which our paper addresses. Note that our application to the large-scale gene expression data demonstrates our method is robust to violation of the model assumptions compared to a random permutation baseline. We make this point in the third paragraph of updated Discussion section.
>
>
> - The Markov blanket assumption concern is important. We admittedly do not have a rigorous explanation for how the violation of this assumption will affect accuracy, generally. The simulation setting in Section 3.2 does attempt to address this concern. The takeaway for these synthetic scenarios is that a neighborhood specification which is the true Markov blanket corresponds to better accuracy compared to when this neighborhood set is estimated. Moreover, the real-data example in Section 3.3.2 shows that although we do not know whether neighborhood sets estimated with non-negative least squares regression are accurate, we can still obtain reasonable results compared to a random permutation baseline. Similarly, Section 3.3.1's application shows we can obtain reasonable results, compared to a Gaussian dag-learning model, when specifying neighborhoods as the most correlated neighbors.
>
>
> - Given that the target parameter of our work is a topological ordering as in Definition 1.1, we believe the accuracy metric in Equation (11) captures concisely a violation of Definition 1.1. If a node with many parents is placed at the start of an ordering before many or all of its parents, then this error metric will penalize proportionally. Likewise, this penalty will occur proportionally if a node with many children nodes is sorted after many or all of its children nodes.
>
> - We do not quite see why Spearman correlation is a better metric for sorting accuracy. Take for example the 3-node DAG, $x_1 \to x_3 \leftarrow x_2$. Two valid topological orderings are $(1,2,3)$ and $(2,1,3)$, yet the Spearman correlation between these two orderings is not perfectly $1$; it is $0.5$ as can be seen when running `cor(c(1,2,3),c(2,1,3),method='spearman')` in `R`. Therefore, if our estimated ordering is correct and we were to use Spearman correlation, we would have to calculate the correlation between the estimated order and all true orders to see that the order is correct. In the worst-case, this comparison could be exponential in the number of nodes. Equation 11 does not require such a one-to-many comparison.

---

### Review · Reviewer_6gVS · 2022-09-13

**Summary Of Contributions:**

The paper studies the problem of learning the topological ordering of the underlying DAG of a structural equation model (SEM) with non-Gaussian additive errors, also known as LiNGAM. The proposed approach learns the topological order by first identifying a root of the DAG and then sequentially appending nodes that would lead to a topological ordering compatible with the true DAG. The appending procedure is based on a likelihood-ratio test. The authors show that in the population setting, the algorithm identifies the true ordering of the DAG. Experiments are conducted to validate mainly the scalability aspect of the proposed algorithm.

**Broader Impact Concerns:**

No broader impact statement is included in the paper. I believe the authors could try addressing how applying this method in practice could have negative ethical implications when wrongly applied or maliciously.

**Requested Changes:**

My following questions/comments are more or less in order. I would appreciate it if the authors could clarify them.

### Major (please address if possible)

1. On page 3, paragraph 2, it is stated that the method runs in $O(pd)$ number of LS updates. It would help to see the **total** computational complexity in terms of $p,d$ for the population case and $p,d,n$ for the finite sample regime, and how it compares to previous work. This is to make a fair comparison against existing work. For instance, it is clear from Algorithm 2, that the overall computational complexity is not $O(pd)$.

1. There is another work by Ghoshal & Honorio, "Learning linear structural equation models in polynomial time and sample complexity," that tackles the linear setting for general error distributions. I suggest also comparing against this work.

1. Why focus on the ordering of the DAG instead of the whole structure? I am aware that if one recovers the ground-truth ordering then discovering the edges becomes a much simpler problem. However, given that the method does not include finite-sample guarantees, I would like to see how the method compares against existing approaches when learning the whole DAG.


**About Experiments**
1. Is there any particular reason why the entries $B_{jk}$ cannot be greater than 1 in absolute value?

1. The scale parameter for each error is somewhat similar. Why not sample from a larger range than $[.4, .7]$? I ask this because it is known that other methods that assume equal variance are robust to small changes in the variances. Since the proposed method does not assume equal variances, I would suggest letting the scale parameter be sampled from a larger range.

1. There should be simulations to corroborate the correctness of the approach in the population setting.

1. I am not sure if the metric in equation 11 is the right one. In particular, given that the graphs are sparse, it is likely that the topological orderings are not unique. For instance, assume we have a naive method that always outputs an empty graph, then any permutation is a valid topological ordering according to Definition 1.1. Thus, what topological ordering is used to compare against the proposed approach? I believe this would make the comparison unfair against other methods that output the whole DAG and can potentially have many valid topological orderings.


### Minor (might strengthen the work)

1. Why not simply write $\hat{\pi}(t) \gets \arg \max_{k \notin \mathcal{A}_t} \mathcal{S}(k,\mathcal{A};\mathbf{X})$ in Algorithm 1 and avoid the unnecessary loop?

1. Is there a typo in Assumption 2.1? In particular, should $g\left(e ; \theta_k\right)=\frac{\theta_0}{\theta_k} g\left(\theta_0 e / \theta_k ; \theta_0\right)$ instead be $g\left(e ; \theta_k\right)=\frac{\theta_k}{\theta_0} g\left(\theta_0 e / \theta_k ; \theta_0\right)$? Otherwise, I am not clear why $\epsilon_k \stackrel{d}{=}\left(\theta_k / \theta_0\right) U$ follows.

1. While three distributions are mentioned that fulfill Assumption 2.1., are there more to add to the list? If so, I could not see any discussion on when it makes sense to assume a particular error distribution.

1. The notation is a bit sloppy in some places. In the definition of LiNGAM, it is not assumed that the error means are zero. However, Assumption 2.1 gives the impression that the errors have mean zero. Also, at the beginning of Page 6, the density $\phi$ is said to have mean zero, however, near the end of page 6, $\kappa$ is defined with a normal distribution with mean $E[X]$, is that not zero?

1. It would help to standardize the notation of the expectations of the likelihood ratios. In eq.(3) the expectation is w.r.t $f_{kt}(r_{kt})$, while in pages 6 and 7, the expectation is w.r.t. $f(x)$.

1. In the definition of $(\mathcal{L}-\tilde{\mathcal{L}})_t (\hat{\pi})$ in Page 7, should it be $ R\_{\hat{\pi}(j)}^{\hat{\pi}} $ instead of $ R\_{\hat{\pi}(j)}^{\pi} $?

1. In Algorithm 2, I think it is more clear to write $S(k, \mathcal{A}_t;\mathbf{R})$ instead of just $S(k;\mathbf{R})$. This is to be consistent with the previous definition of $S$.




**Strengths And Weaknesses:**

### Strengths
* The exposition of the problem setting and the main ideas of the proposed approach are clear.
* The algorithm is easy to digest and leads to discovering the true ordering of the underlying DAG in the population case.
* The approach is flexible enough to incorporate prior knowledge about the structure of the DAG.
* The method can scale to high dimensions with competitive runtimes.

### Weaknesses
* There are some details about the method that I found unclear. I have a few questions that I will elaborate on below.
* The sample complexity of the method is unclear and not discussed at all.
* The experimental setting does not include the population case to corroborate the theoretical results. Moreover, I found part of the experimental setting somewhat strange; see below.
* The approach would work only for linear models, which is a significant limitation for real-world problems.

---

> ### Author Response · Authors · 2022-09-30
> **Review of Paper313 by Reviewer 6gVS**
>
> We agree with all of the minor changes and typo corrections that would make the paper better. We thank the reviewer for taking the time to point out these points of improvement.
>
>
>
> - We will include a broader impact statement as requested. Please let us know whether the following addresses the concern about the broader impact of this paper: "Our scalable causal discovery method comes with extra modeling assumptions, not to mention the fact that we are making use of the causal sufficiency--no unobserved confounder or selection bias--assumption. Therefore, it is important for a responsible practitioner to not over-extend their conclusions with this or related data mining methods.''
>
>
>
> - Regarding the experimental settings addressing the correctness of population setting (Theorem 2.5), we feel the largest n simulation settings in Section 3.1's Figure 2 address the concern: note the accuracy increase with larger $n$.
>
> - Regarding sample complexity, is this term taken to mean statistical convergence rate, or is it taken to mean the computational complexity of Algorithm 2 for a finite $n$?
>
>
> - The second to last paragraph of Section 2.4 now states the computational complexity of Algorithm 2 as $\mathcal{O}(pdmn)$
> in the number of scalar to scalar multiplication, division, addition, or difference calculations. Here, $m$ is the maximum number of ancestors a node can have, which is a bound on the iterations for the inner-most for loop in Algorithm 2.
>
> - We do not believe that a comparison to Ghoshal & Honorio (2017) is fair for either model estimation strategies. Ghoshal \& Honorio (2017) works under a bounded variance assumption (see their Assumption 1), whereas our theoretical results do not require a bounded variance assumption. If one wants to specify a linear SEM with bounded variances, then one should use the Ghoshal \& Honorio (2017) or other specialized methods. If one wants to specify a Linear SEM with non-Guassian errors satisfying Assumption 2.1-2.3 in our paper, then one should apply our method. The specific choice of simulation parameters, namely the weighted adjacency matrix entries and scale parameters, are so that the marginal variance of $X_j$ does not explode when $j$ is a node with many ancestors.
>
> - Our motivation for learning a topological ordering is that it is a relatively simpler problem to tackle compared to searching across the super-exponentially large space of DAG structures. Under Assumption 2.3 and regardless of whether the order estimate satisfies Definition 1.1, a graph output by our Algorithm 2 will have false positive edges corresponding to at least the number of times two non-adjacent nodes have a common child in the true graph. This is because a Markov blanket contains all co-parents (or "spouses"). Moreover, the number of incorrectly oriented true edges will be the total number of true edges times the accuracy metric in Equation (11). Thus, to produce a good estimated graph given our estimated sort, one needs an additional step to apply a sparse regression of each $X_j$ on the variables sorted in front of it to estimate the support (parent set). This is the well-studied support recovery problem in sparse linear regression. We make this point clear in the second paragraph of the updated Discussion section.
>
> - Given that the target parameter of our work is a topological ordering as in Definition 1.1, we believe the accuracy metric in Equation (11) captures concisely a violation of Definition 1.1. It only takes into account all child-parent pairs in the underlying graph, not the number of topological orderings which are not unique. If a node with many parents is placed at the start of an ordering before many or all of its parents, then this error metric will penalize proportionally. Likewise, this penalty will occur proportionally if a node with many children nodes is sorted after many or all of its children nodes.
>
>
> - We note that Assumption 2.1 is correct as presently stated. Note that if $U=aV$, then the CDF of $U$ is $Pr(U\leq u)=Pr(V\leq u/a)$. Taking the derivative of the CDF with respect to little $u$ to obtain the PDF of $U$ gives $g(u/a)/a$, where $g$ is the PDF of $V$.

---

### Review · Reviewer_BwK3 · 2022-09-16

**Summary Of Contributions:**

This work considers the problem of estimating a topological ordering of a DAG. A sequential procedure greedily adds nodes to an ordering based on a novel likelihood ratio score. The authors show at the population level, with assumptions of a LiNGAM (Linear Non-Gaussian Additive Model) holding, a true ordering will be identified. This framework allows prior knowledge of Markov blankets of nodes to improve computational efficiency. Synthetic and real data experiments show that this method is tractable for larger graphs where existing methods are not. Overall, the paper is well written, and care is taken by the authors to articulate the proposed method. The empirical evaluation answer most, but not all, of the questions I have regarding the performance of the method.

**Requested Changes:**

Requested changes:

- The first sentence of the first paragraph of Section 1.2 is missing an “is”.

- Figure 2 may be trying to summarize too many results at once. I would recommend separating the cases of correctly specified error distributions from the misspecified cases. The first figure will show that the method accomplishes the goal it is theoretically set up to accomplish, while the second figure will show that it is robust to a potential misspecification, which do not need to be presented simultaneously. It also may be worth sorting the dataset by p, rather than alphabetically. Currently, it is a bit difficult to process all the information being presented at once.

Suggested changes to strengthen work:

- The simulation studies utilize a portion of the data to create prior knowledge of the Markov blanket, and it is clearly stated in that section that a consistent method should be used for estimation. However, earlier in the paper, it is suggested that the prior information may be available through domain knowledge, which is susceptible to being incorrect. How robust is this method to incorrect specifications? A brief simulation study, perhaps better suited for an appendix, would be worthwhile to have. The presented evidence show that correct prior knowledge can improve computation

**Strengths And Weaknesses:**

Strengths:

- The proposed method offers a tractable solution that can be improved with prior knowledge.

- The main contribution is presented well. The proposed likelihood ratio score is natural. The progression of theorem 2.5, lemma 2.6, and lemma 2.7 is easy to follow. The CLT argument following lemma 2.7 provided useful intuition.

- The experimental setup demonstrates the ability of the method to scale where existing methods do not. In general, the empirical evidence supports the main claims.

Weaknesses

- The presentation of the empirical results could be improved (see requests and suggestions). The empirical results shown are based on a limited number of repetitions (30, 50, 90).

- Evaluation of misspecified prior knowledge (Markov blanket) not presented.

---

> ### Author Response · Authors · 2022-09-30
> **Review of Paper313 by Reviewer BwK3**
>
> We thank the reviewer for taking the time to suggest edits that will strengthen the readability of this paper.
>
> - We agree with the requested changes, especially that splitting the former Figure 2 in Section 3.1 gives better readability. Please see the new Figure 2 and Figure 3 in the updated manuscript.
>
> - We agree that the Markov blanket assumption concern is important. We admittedly do not have a rigorous explanation for how the violation of this assumption will affect accuracy, generally. The simulation setting in Section 3.2 does attempt to address this concern. The takeaway for these synthetic scenarios is that a neighborhood specification which is the true Markov blanket corresponds to better accuracy compared to when this neighborhood set is estimated. Moreover, the real-data example summarized in Section 3.3.2's Figure 7 shows that although we do not know whether neighborhood sets estimated with non-negative least squares regression are accurate, we can still obtain reasonable results compared to a random permutation baseline. Similarly, the application in Section 3.3.1 summarized in Figure 6 shows we can obtain reasonable results, compared to a Gaussian dag-learning model, when specifying neighborhoods as the most correlated neighbors.
>
> - Regarding the replication amount for the empirical results in Section 3.1, we do make an attempt to make more comparisons by providing results for several DAG architectures. We do not have a larger number of replicates for each individual synthethic setting because the competing methods do not scale as well.

---

### Review · Reviewer_ZRRu · 2022-09-16

**Summary Of Contributions:**

In this work, the authors propose an approach for learning variable (or node) orderings, according to LiNGAM assumptions, by iteratively adding one new variable to the order. The variable added at each iteration is selected based on a likelihood ratio between a non-Gaussian and Gaussian term. Under a given set of assumptions, the procedure guarantees to find an ordering consistent with the true structure. A finite sample version of the approach is also given, which largely entails fitting (and efficiently updating) regression models. A set of empirical experiments suggest that the proposed approach is competitive with SOTA in terms of accuracy and has better computational scalability.

**Broader Impact Concerns:**

No major ethical concerns

**Requested Changes:**

Critical changes

* Add an appendix (table, etc.) that contains the major notation used in the main text
* Add the processed data files and brief documentation for running the experiments in the text
* Clarify earlier about “valid” vs. “invalid” nodes
* Perform analysis similar to that in Figure 4 for both HighDimLiNGAM  and ScoreLiNGAM for a few settings and datasets, such as small-data settings for alarm and win95pts
* Address the various points about the plots, etc., mentioned above
* Properly format the references, including proper capitalization, etc.
* Address the various typos mentioned above

Suggested changes
* Package the source code as a proper R package (“DESCRIPTION”, “R” folder, etc.)
* Clarify about the requirement on the same distribution family, especially if it reveals something interesting about the approach
* Run HighDimLiNGAM on the unscaled data, for at least a few settings to confirm that the observed trends are not due to scaling
* Address the observation that HighDimLiNGAM does not improve with more data, but the proposed approach (and DirectLiNGAM) do improve as more data is available.
Address the observation that DirectLiNGAM is typically more accurate than ScoreLiNGAM despite not being given the MB
* Show a few examples of learned causal relations which have wet lab support (from STRING, etc.)


**Strengths And Weaknesses:**

While I am not an expert in this area, I generally found the paper to be well-written. For example, I found it very helpful when the authors provided text descriptions and inutions on what the various equations represented (though see below for some specific questions). However, due to the dense notation required for this topic, I believe the readability would be significantly improved by adding an appendix, table, etc., which lists the notation used in the main text.

The reproducibility of the work is somewhat limited. While the authors do provide an R notebook for running some small examples, it does not follow typical R package structure. More importantly, the simulated and processed data files are not available. Thus, it is not possible to reproduce the results.

As a non-expert, I did not verify the proofs in detail. However, they seemed reasonable to me. However, I do have several technical questions.

* Do the entries of the error vector need to follow the same distribution family? For example, could one term follow a Laplacian and another follow a logistic? or even Gaussian? Do the theoretical results depend on them all following distributions in the same family?
* What is meant by a “valid” vs. an “invalid” node? From Lemma 2.6, this seems to refer to a node consistent with the true topological order (in the population setting). The term is used several times before that, though, so it should at least be defined clearly before use.

The empirical evaluation is modest. A strength is that the authors investigate the case when the distributional family is mis-specified. However, the comparison to HighDimLiNGAM can be improved by using an experimental design similar to that in Figure 4, in which both methods are given an empirically determined subset of candidates for the MB, rather than the ground truth. Also, since HighDimLiNGAM is not invariant to re-scaling, it would be meaningful to include results on it with the unscaled data.

A major limitation of the empirical analysis is that it is not very deep. For example, the accuracy of HighDimLiNGAM does not seem to particularly depend on the size of the dataset; on the other hand, the proposed approach generally improves as more data is available. It would be insightful to the differences which lead to this behavior. Similarly, it is striking that DirectLiNGAM does not require the MB as input, yet it typically outperforms ScoreLiNGAM in terms of accuracy; I appreciate that the scalability is an important aspect of the proposed approach (and that DirectLiNGAM is not an option when n <= p), but it would still be useful to point to specific differences in the methods that lead to these results.

I also have a few specific suggestions about presenting the results
* I may have missed it, but I do not believe it is ever stated what the box plots are showing (quartiles, standard deviation, variance, standard error, etc.)
* Sorting Figure 2 by number of variables rather than alphabetically
* Including the density of the networks, in terms of number of edges compared to the number of possible edges (or similar)
* Show standard deviation or similar in Figure 3 (though it may be very low)
* Split Figure 6 into multiple plots or similar; especially since the y-axis values change so much from part to part, it is very difficult to follow. Also, use similar formatting (i.e., filled-in boxes) as the other box plots.

The gene expression analysis would be improved with some semi-qualitative analysis comparing the learned relationships (and strengths) with wet lab-validated interaction data. For example, the STRING database (https://string-db.org/) is one open resource which includes such interaction data.

Typos, etc.
* The references are not consistently formatted.
* The second sentence of Section 1 is a run-on sentence.
* The first sentence in Section 1.2 is missing a verb.
* When referring to specific “Definitions”, “Assumptions”, etc., the first letter is always capitalized, but it is never capitalized when referring to specific “equations.”

---

> ### Author Response · Authors · 2022-09-30
> **Review of Paper313 by Reviewer ZRRu**
>
> We generally agree with the critical requested changes and thank the reviewer for taking the time to suggest edits that will strengthen the readability of this paper.
>
> - Section 2.1 now contains a summary of our notation.
>
> - The start of Section 2.3 clarifies our meaning of "valid" and "invalid" nodes before the subsequent use of these words.
>
> - The first paragraph of Section 3 clarifies all figures with boxplots.
>
> - Regarding the observation that the HighDimLiNGAM method does not improve as sample size increases when the data matrix is re-scaled, we note that it could may very well be that much more data is needed for it to work well. We agree that it is striking that DirectLiNGAM does not require the MB as input, yet it performs well in terms of accuracy. We point these phenomena out in the last two sentences of paragraph 5 in Section 3.1.
>
> - The concern about whether the error vector's entries need to follow the same distribution family is an important one. Our answer is that is not clear how our current theoretical results, making use of information theoretic quantities, will come about without this assumption. Extensions of our work may like to explore whether this assumption can be relaxed. This relaxation is suggested by our simulation results where the density family is mis-specified. Such an extension may require that any two error terms' differential entropy not be too dissimilar after re-scaling the error terms to have unit variance. This is now addressed in the third paragraph of the Discussion section.
>
>
> - We have added the simulation files and brief documentation for running the experiments in the text.
>
> - We plan to package the source code as a proper R package on github.
>
> - We plan to add to the simulations in Section 3.1 to include the case where the neighborhood sets are estimated from data, too.
>
> - We plan to re-do simulations in an update of the manuscript to include the case that we do not re-scale the data matrix.
>
>
>
> - We will explore improving the gene expression analysis by comparing the learned relationships (and strengths) with wet lab-validated interaction data. We thank the reviewer for the STRING database suggestion. However, we note that we authors lack very strong wet-lab/domain knowledge, which is why we refrained from any causal claims and included a discussion about hidden vs. observed gene expressions in the conclusion of the paper. Validation of the learned structural equation model for the gene expression data is currently done by calculating performance on a test dataset.

---

### Decision · Action_Editors · 2022-11-03

**Recommendation:** Accept with minor revision

**Comment:**

The authors have substantially revised the paper in response to reviewer comments. They have listed a few changes that are still pending. They should also review the claim of robustness to misspecification and identify ways in which it can be better aligned with their evidence if possible.

**Audience:**

The problem of graph structure learning in causal inference is important and algorithms for learning the topological ordering of a DAG have widespread applications. Furthermore, theoretical and methodological research on this topic are of interest to many researchers in the field.

**Claims And Evidence:**

The reviewers have thoroughly reviewed the claims and evidence. They are generally in agreement that the claims are well-supported by the evidence, though there are some concerns that claims regarding robustness to misspecification may be over-reaching.